# Dosage compensation can buffer copy-number variation in wild yeast

**James Hose[1], Chris Mun Yong[1], Maria Sardi[1], Zhishi Wang[2], Michael A Newton[2,3], Audrey P Gasch[1,3]\***

[1]Laboratory of Genetics, University of Wisconsin-Madison, Madison, United States; [2]Department of Statistics, University of Wisconsin-Madison, Madison, United States; [3]Genome Center of Wisconsin, University of Wisconsin-Madison, Madison, United States

**Abstract** Aneuploidy is linked to myriad diseases but also facilitates organismal evolution. It remains unclear how cells overcome the deleterious effects of aneuploidy until new phenotypes evolve. Although laboratory strains are extremely sensitive to aneuploidy, we show here that aneuploidy is common in wild yeast isolates, which show lower-than-expected expression at many amplified genes. We generated diploid strain panels in which cells carried two, three, or four copies of the affected chromosomes, to show that gene-dosage compensation functions at >30% of amplified genes. Genes subject to dosage compensation are under higher expression constraint in wild populations—but they show elevated rates of gene amplification, suggesting that copy-number variation is buffered at these genes. We find that aneuploidy provides a clear ecological advantage to oak strain YPS1009, by amplifying a causal gene that escapes dosage compensation. Our work presents a model in which dosage compensation buffers gene amplification through aneuploidy to provide a natural, but likely transient, route to rapid phenotypic evolution.

\*For correspondence: agasch@wisc.edu

**Reviewing editor**: Duncan T Odom, Cancer Research UK Cambridge Institute, United Kingdom

## Introduction

Susumu Ohno proposed over 40 years ago that gene duplication could provide a major force in the evolution of new gene functions, by relaxing constraint on gene sequences and allowing one or both gene copies to evolve (*Ohno, 1970*). The genomic era has largely borne out that hypothesis, and many studies have characterized the outcomes of whole and partial genome amplification (*Jaillon et al., 2009*). The immediate consequence of duplication is assumed to be increased expression of the affected genes, and in some cases the increased expression provides a selective advantage (e.g., *Sandegren and Andersson, 2009*; *Chang et al., 2013*; *Edi et al., 2014*). Over longer periods, the relaxed constraint afforded by functional redundancy allows one or both gene copies to evolve (*Ohno, 1970*), driving sub- and neo-functionalization (*Lynch and Force, 2000*; *Lynch et al., 2001*), expression divergence (*Gu et al., 2004*, *2005*; *Li et al., 2005*; *Wang et al., 2012*), and network rewiring (*Presser et al., 2008*; *Freschi et al., 2011*; *De Smet and Van de Peer, 2012*).

Whole and partial chromosome amplification through aneuploidy is frequently observed in laboratory evolution studies and in drug-resistant fungal pathogens (*Hughes et al., 2000*; *Dunham et al., 2002*; *Selmecki et al., 2006*; *Pavelka et al., 2010*; *Yona et al., 2012*; *Ni et al., 2013*), suggesting that aneuploidy is a rapid route to phenotypic evolution. However, aneuploidy comes with a fitness cost, most famously in cases of human aneuploidies such as Down syndrome (DS) (*Antonarakis et al., 2004*). The reasons for aneuploidy toxicity are not entirely clear but may be due to increased expression from genes that are toxic when overexpressed (*Siegel and Amon, 2012*). Several studies have used *Saccharomyces cerevisiae* as a model for aneuploidy syndromes, since laboratory strains are extremely sensitive to chromosomal amplification. Laboratory strains with

**eLife digest** Evolution is driven by changes to the genes and other genetic information found in the DNA of an organism. These changes might, for example, alter the physical characteristics of the organism, or change how efficiently crucial tasks are carried out inside cells. Whatever the change, if it makes it easier for the organism to survive and reproduce, it is more likely to be passed on to future generations.

DNA is organized inside cells in structures called chromosomes. Most of the cells in animals, plants, and fungi contain two copies of each chromosome. However, sometimes mistakes happen during cell division and extra copies of a chromosome—and hence the genes contained within it—may end up in a cell. These extra copies of genes might help to speed up the rate at which a species evolves, as the 'spare' copies are free to adapt to new roles. However, having extra copies of genes can also often be harmful, and in humans can cause genetic disorders such as Down syndrome.

In the laboratory, chromosomes are commonly studied in a species of yeast called *Saccharomyces cerevisiae*. This species consists of several groups—or strains—that are genetically distinct from each other. Over the years, breeding the yeast for experiments has created laboratory strains that have lost some of the characteristics seen in wild strains. Earlier studies suggested that these cells fail to grow properly if they contain extra copies of chromosomes. Now, Hose et al. have studied nearly 50 wild strains of *Saccharomyces cerevisiae*. In these, extra copies of chromosomes are commonplace, and seemingly have no detrimental effect on growth. Instead, Hose et al. found that cells with too many copies of a gene use many of those genes less often than would be expected. This process is known as 'dosage compensation'. This dosage compensation has not been observed in laboratory strains, in part because the extra gene copies make them sickly and hard to study.

Together, the results provide examples of how dosage compensation could help new traits to evolve in a species by reducing the negative effects of duplicated genes. This knowledge may have broad application, from suggesting methods to alleviate human disorders to implicating new ways to engineer useful traits in yeast and other microbes.

forced aneuploidy are extremely slow growing, regardless of the chromosome amplified (*Torres et al., 2007*; *Pavelka et al., 2010*). Transcriptomic and proteomic studies in these strains reported proportionately higher expression from virtually all amplified genes (*Torres et al., 2007*; *Pavelka et al., 2010*; *Torres et al., 2010*; *Sheltzer et al., 2012*), with a handful of exceptions recently identified at the protein level (*Dephoure et al., 2014*). The apparent lack of dosage compensation is consistent with another study by *Springer et al. (2010)*, which found that expression at hemizygous genes is not up-regulated to compensate for reduced gene copy. While these studies have generated important results on aneuploidy intolerance in these particular strains, one caveat is that they were all done in laboratory strains, which have lost many features inherent in wild strains (*Kvitek et al., 2008*; *Lewis et al., 2010*; *Lewis and Gasch, 2012*). A remaining question is the extent to which aneuploidy occurs in nature and contributes to phenotypic variation in the wild.

Here, we report that chromosomal amplification is common in non-laboratory yeast strains, which are inherently tolerant of aneuploidy and display an active mode of gene-dosage compensation at the transcript level, for specific classes of amplified genes. Strikingly, genes subject to dosage compensation are buffered against copy-number variation (CNV) and thus show elevated rates of gene amplification in natural isolates. Our results raise new implications for the role of aneuploidy in phenotypic evolution and the mechanisms cells use to tolerate it.

## Results

We sequenced the genomes of 47 non-laboratory yeast strains, including wild, clinical, and industrial isolates (*Supplementary file 1*) and assessed the copy number of the 16 yeast chromosomes. Nearly a third of all strains carried whole (12 strains) or partial (2 strains) chromosome amplification (*Figure 1A*). Three strains harbored multiple aneuploidies, the extreme being insect-associated strain Y2189 that amplified four different chromosomes. Some chromosomes were amplified in multiple unrelated strains: Chromosome III (Chr 3), Chr 9, and Chr 12 were each amplified in different sets of

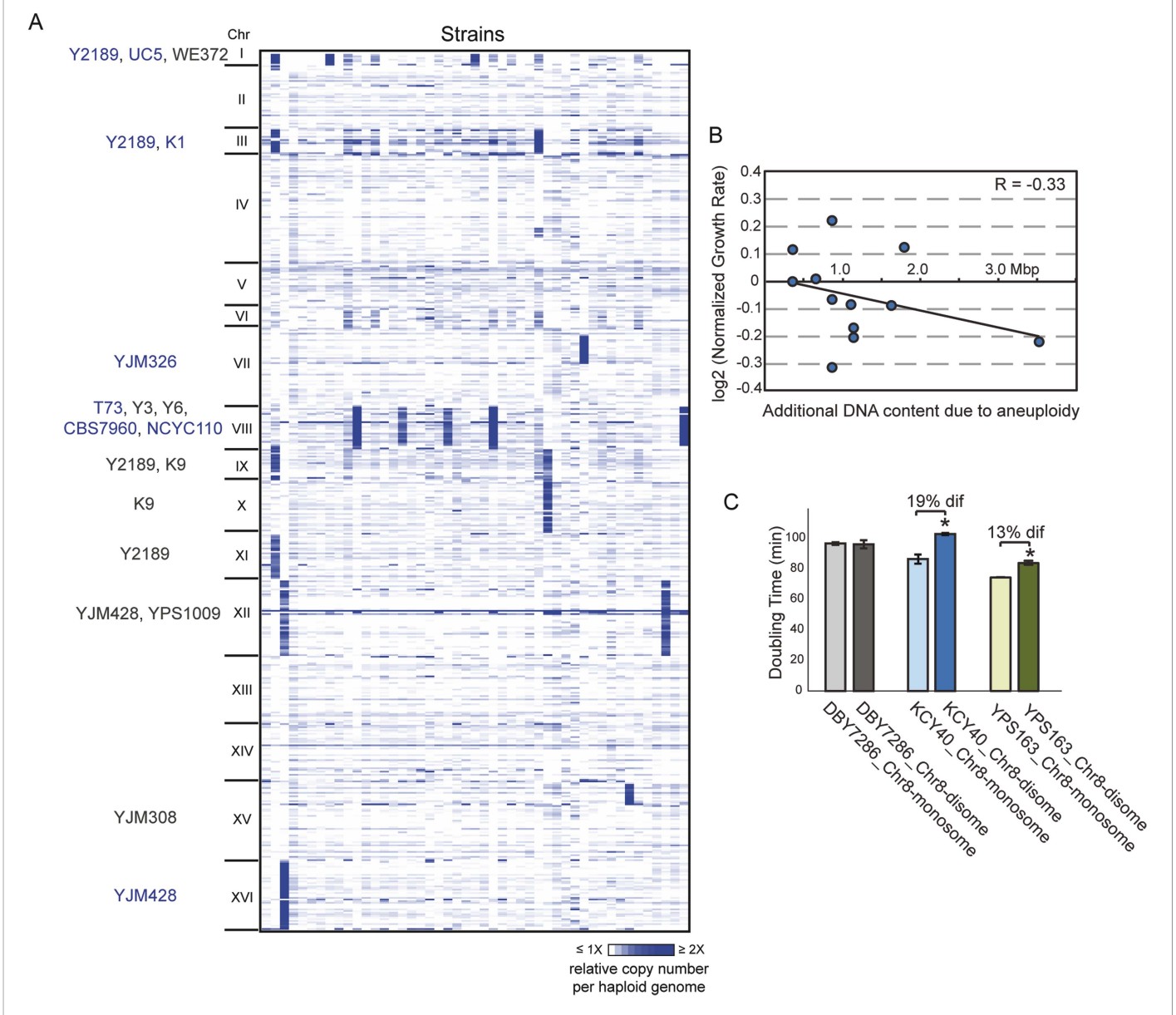

**Figure 1**. Aneuploidy is common in non-laboratory strains. (**A**) Relative DNA copy ($\log_2$ (RPKM)) per gene (rows) across each of the 16 chromosomes for 47 sequenced strains (columns). Strains with 1.5× (gray text) or 2× (blue text) chromosomal copy per haploid genome are annotated by name. (**B**) Growth rates of aneuploid strains normalized to the niche-specific growth rate (see 'Materials and methods') plotted against the additional DNA content in each strain. (**C**) Haploid strains with a duplication of Chr 8 (labeled as 'disome') were selected from euploid parents (labeled as 'monosome') including the S288c-derived DBY7286, derived vineyard strain KCY40, and a haploid derivative of oak strain YPS163, see 'Materials and methods'. Doubling times in YPD medium represent the average of four biological replicates; the average difference ('dif') in growth rate is indicated. An asterisk represents statistically significant differences in doubling time (p < 6e-4, T-test).

strain pairs, Chr 1 (the smallest yeast chromosome) was amplified in three strains, and Chr 8 was aneuploid in five unrelated isolates. The higher incidence of Chr 8 amplification may reflect a higher frequency of mitotic error, since diploid mutation-accumulation (MA) lines propagated in the near-absence of selection amplify Chr 8 at a higher rate (*Zhu et al., 2014*).

However, several lines of evidence refute the model that these aneuploidies represent deleterious mutations yet to be removed by selection. Unlike diploid MA lines that were at maximum trisomic for specific chromosomes, half of naturally aneuploid strains are tetrasomic (*Figure 1A*). Furthermore, natural

aneuploids showed no significant growth reduction compared to closely related euploid reference strains (p = 0.19, paired T-test), although there was a slight negative correlation between growth rate and extra DNA content (R = −0.3, *Figure 1B*). This is in stark contrast to aneuploid laboratory strains, which show extreme growth retardation (*Torres et al., 2007*; *Pavelka et al., 2010* and below). We also found that chromosomal amplification was stable for >200–400 generations in four interrogated wild strains, whereas the W303 laboratory strain generally loses aneuploidy within 20 generations. To distinguish if the tolerant strains have adapted to aneuploidy or if non-W303 strains can inherently accommodate chromosome amplification, we selected aneuploid derivatives of several naturally euploid parents (see 'Materials and methods'). We found little to no growth defect in derived aneuploid strains (*Figure 1C*). Thus, *S. cerevisiae* isolates are inherently tolerant of chromosomal amplification, which is common in nature.

## A common aneuploidy response is distinct from that in the laboratory strain

Aneuploid laboratory strains are reported to show proportionately higher expression from virtually all amplified genes, causing proteotoxicity from excess protein production (*Torres et al., 2007*; *Pavelka et al., 2010*; *Torres et al., 2010*). We therefore investigated transcriptome profiles through RNA deep sequencing (RNA-seq) in six naturally aneuploid strains normalized to paired euploid reference strains that are closely related (thereby minimizing neutral expression differences unrelated to aneuploidy, see 'Materials and methods').

Consistent with their near-normal growth rates, naturally aneuploid strains did not activate the environmental stress response as seen in sickly laboratory aneuploids (*Torres et al., 2007*; *Sheltzer et al., 2012*) (*Figure 2A*). However, we detected a weak signature common to several aneuploid strains, including the up-regulation of 69 unamplified genes (enriched for oxidoreductases) and reduced expression of 269 unamplified genes (strongly enriched for mitochondrial ribosomal protein (RP) genes and genes involved in respiration) in at least three of the six aneuploid strains (*Figure 2B*). We tested the respiratory capabilities of naturally aneuploid yeast with variable chromosome copy number and found no growth defect on non-fermentable carbon sources (*Figure 3*). In contrast, a diploid W303 strain trisomic for Chr 12 ('W303_Chr12-3n') displayed an exacerbated growth defect on non-fermentable carbon sources, and we were unable to make the tetrasomic W303_Chr12-4n strain that retained its mitochondrial genome, despite numerous backcrossing attempts (*Figure 3*). This suggests that differences in mitochondrial function may contribute to differences in aneuploid tolerance across strain backgrounds. Interestingly, up-regulation of oxidoreductases and down-regulation of mitochondrial genes are a hallmark of DS (*Conti et al., 2007*; *Lintas et al., 2012*; *Helguera et al., 2013*; *Valenti et al., 2014*) (see Discussion).

## Many amplified genes display lower-than-expected expression

Given the significant phenotypic differences in laboratory vs non-laboratory aneuploid strains, we were particularly interested in the expression of amplified genes. We investigated transcript abundance relative to DNA copy for amplified genes, interrogating 2204 genes spanning eight amplified chromosomes across the six aneuploids. Across all strains, nearly 40% of amplified genes showed lower expression per gene copy compared to the paired euploid (*Figure 4A*, blue points, see 'Materials and methods'). These were enriched for genes encoding RPs, translation factors, proteins localized to the nucleus or to mitochondria, and other groups (p < 1e-5, *Supplementary file 2*). The lower-than-expected expression could not be explained by a general repression response to the aneuploidy, since only 39 of the 838 affected genes were part of the common response described above. In contrast, amplified genes with expression proportionate to gene copy (*Figure 4A*, gray points) showed distinct enrichment for genes encoding proteins localized to the cytoplasm or to membranes, stress defense proteins, and kinases and transferases (p < 1e-5, *Supplementary file 2*). A subset of genes was expressed ≥1.5× higher than expected per gene copy (*Figure 4A*, magenta points), and these were enriched for genes that influence morphology (p = 1e-5, see more below).

## Reduced expression in isogenic strain pairs implicates dosage compensation

Two models could explain the reduced expression from amplified genes. New mutations, through adaptation or drift, could heritably reduce expression at toxic amplified genes. Alternatively, wild strains may actively down-regulate expression in proportion to gene dose, known as dosage

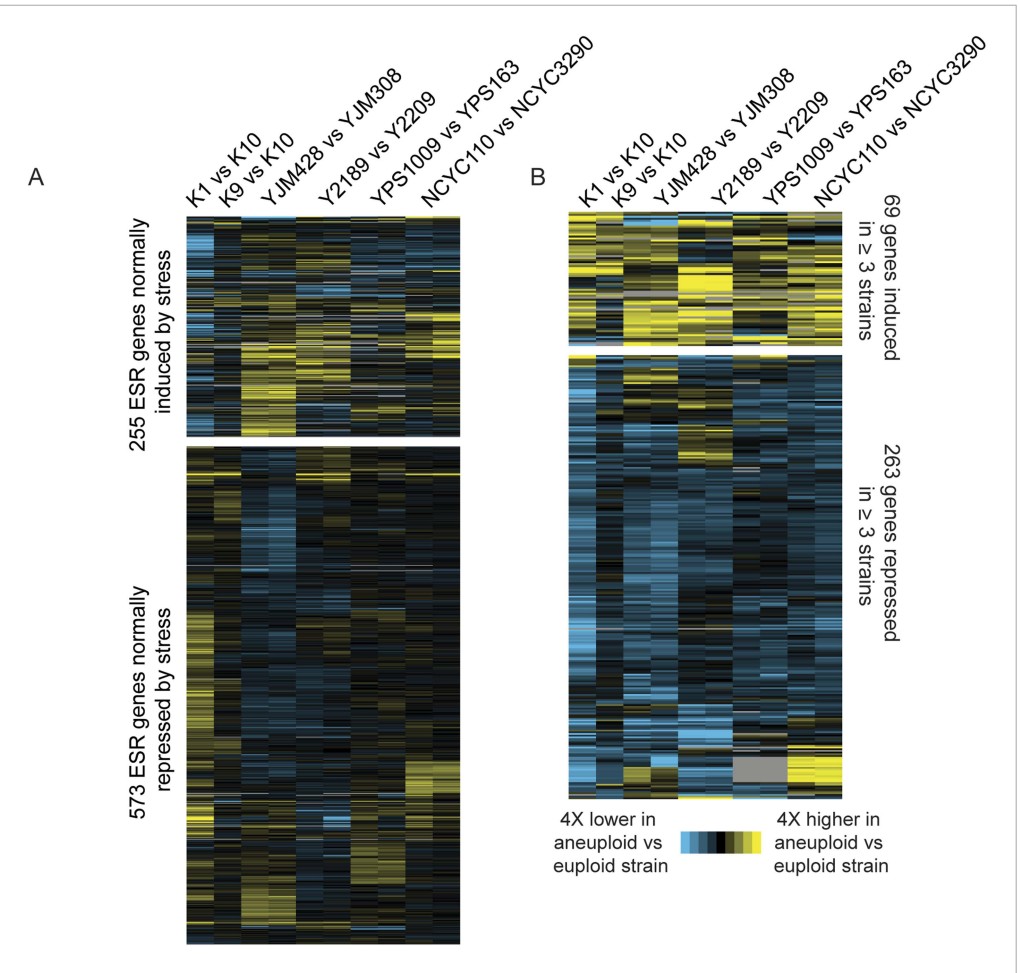

**Figure 2**. Naturally aneuploid strains show a weak common response to aneuploidy but no activation of the ESR.
**A**) Expression of genes in the yeast environmental stress response (ESR) in aneuploid vs genetically related euploid strains. Biological replicates are shown for YJM428, Y2189, YPS1009, and NCYC110. The magnitude of the expression difference ($\log_2$ fold change) is as indicated in the key. **B**) Expression of 69 genes with higher expression (top) and 263 genes with lower expression (bottom) in aneuploid strains vs their paired euploid control.

compensation. To distinguish between these possibilities, we generated isogenic aneuploid–euploid strain pairs for three wild strains. We isolated euploid derivatives of strain T73_Chr8-4n (denoting four copies of Chr 8 in the diploid strain) and YJM428_Chr16-4n by serial passaging for many generations. We also used drug-based selection to isolate a mutant of naturally euploid oak-soil strain YPS163 in which Chr 8 was amplified (see 'Materials and methods'). We then conducted duplicate RNA sequencing (RNA-seq) analysis in the isogenic aneuploid–euploid pairs and identified genes on Chr 8 or Chr 16 with lower-than-expected expression, as above. Because the strains are nominally isogenic (see 'Materials and methods'), most expression differences between the strain pairs are an active response to the aneuploidy.

Recapitulating the results shown above, roughly 11–36% of amplified genes, depending on the strain, showed lower-than-expected expression (*Figure 4B*, blue points), while 2–4% showed higher-than-expected expression (*Figure 4B*, magenta points). Few of these genes participate in the common response to chromosome amplification, suggesting a mechanism of dosage compensation. A substantial fraction of the genes scored in the forced YPS163 aneuploid displayed reduced expression, even though this strain had little time to adapt to the aneuploidy. Thus, dosage compensation is likely an inherent trait in *S. cerevisiae* that functions at a subset of yeast genes.

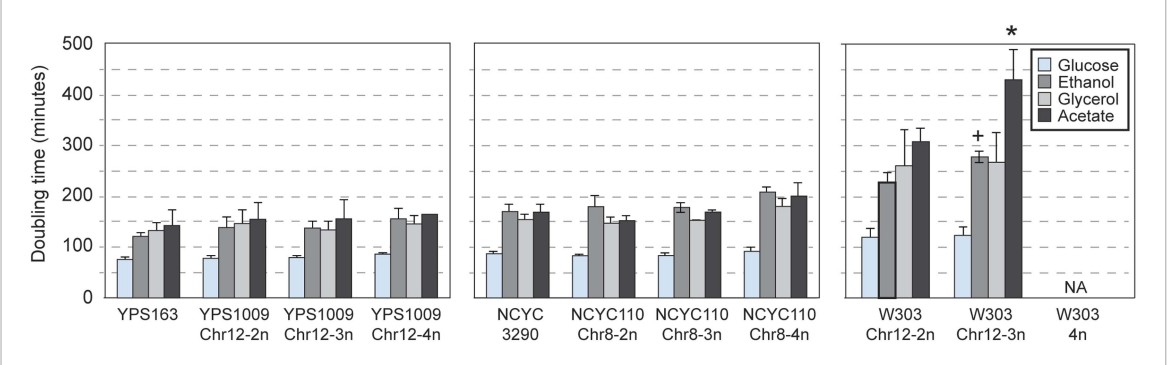

**Figure 3**. Naturally aneuploid strains have no respiratory defect. The average and standard deviation of doubling times measured for strains growing in yeast extract-peptone medium supplemented with 2% glucose (fermentable), 2% ethanol, 3% glycerol, or 2% acetate (non-fermentable) is shown for indicated strains. Isogenic, diploid panels of YPS1009, NCYC110, and W303 carried variable copy of Chr 12 or Chr 8 (see text). The relative growth rate on non-fermentable carbon sources (normalized to each strain's growth rate on glucose) was not significantly different for most aneuploid strains, with the exception of the W303_Chr12-3n strain that showed a greater growth defect when grown on ethanol or acetate compared to glucose (+ = $p < 0.07$, * = $p < 0.05$, paired T-test). The W303_Chr12-4n did not grow on non-fermentable carbon sources.

## Gene classification further refines expression patterns at amplified genes

To more accurately define genes subject to dosage compensation vs heritable polymorphisms, we next generated isogenic strain panels for oak strain YPS1009 and West African strain NCYC110, in which isogenic diploids carry two, three, or four copies of Chr 12 or Chr 8, respectively (*Figure 5A*). Expression in each strain within the YPS1009_Chr12 or NCYC110_Chr8 panels was normalized to

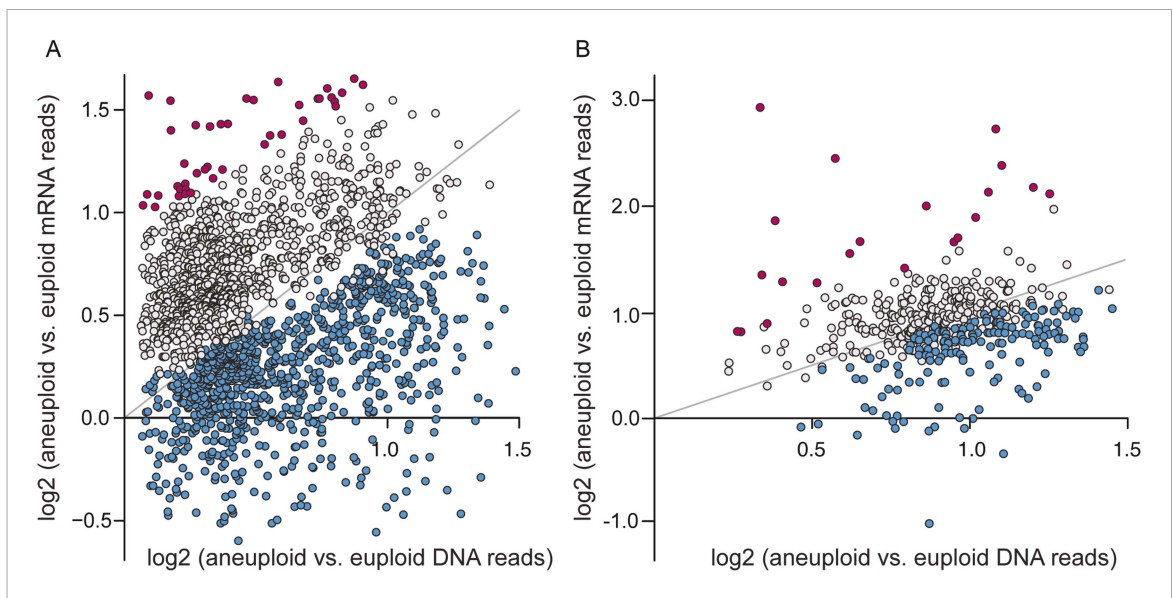

**Figure 4**. A large fraction of amplified genes are expressed lower than expected in aneuploid isolates. The average $\log_2$(aneuploid vs euploid RNA-seq reads) per amplified gene was plotted against average $\log_2$(aneuploid vs euploid DNA seq reads) measured for that gene, in each of the interrogated aneuploid strains. (**A**) The combined set of amplified genes measured in each wild isolate normalized to a genetically related euploid reference strain, and (**B**) Amplified genes measured in two tetrasomic diploid strains and one disomic haploid strain normalized to isogenic euploids. Genes with lower-than-expected expression are plotted in blue and genes with higher-than-expected expression are plotted in magenta (see 'Materials and methods'). The expected relationship representing proportionate increases in expression (slope of 1.0, intercept of 0) is shown in each panel.

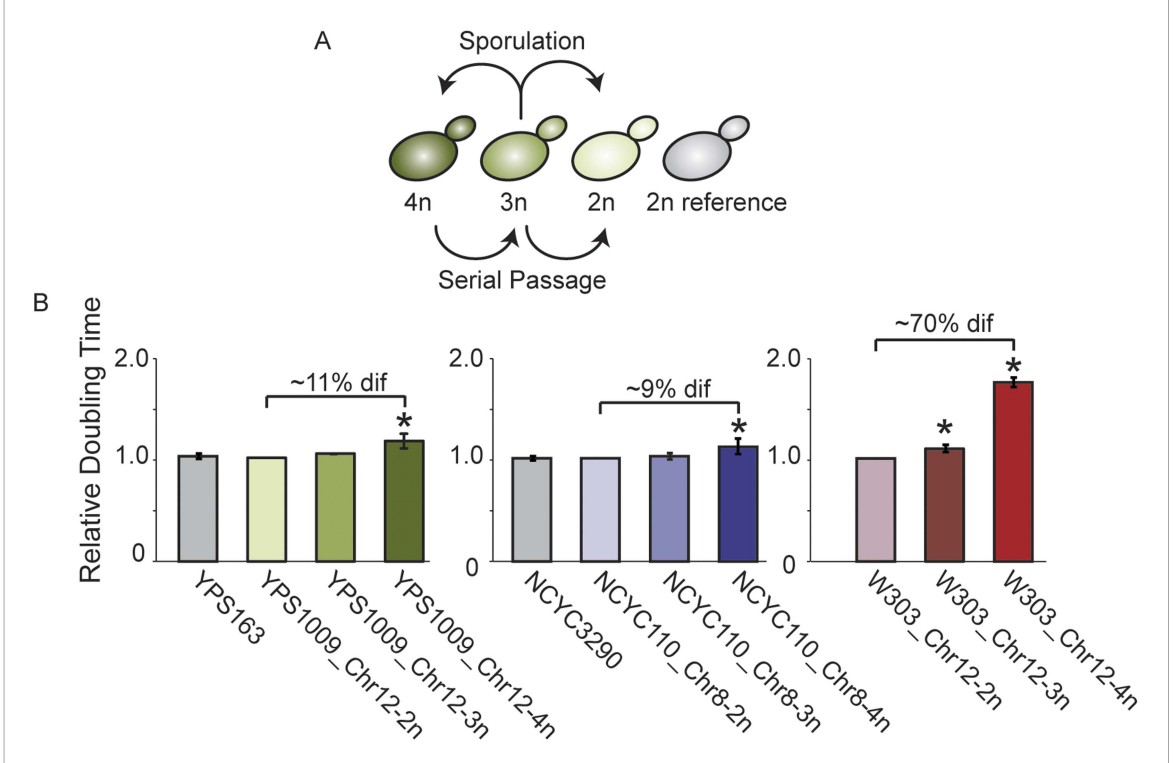

**Figure 5**. Aneuploid strain panels. (**A**) Strain panels were generated by sporulating YPS1009 (naturally trisomic for Chr 12) or serial passaging of NCYC110 (naturally tetrasomic for Chr 8). (**B**) Average and standard deviation of doubling times for strains in each panel and euploid reference strains YPS163 or NCYC3290, normalized to the respective 2n strain in each panel. Asterisks represent significant differences (p < 0.01).

a closely related euploid reference, YPS163 or NCYC3290, respectively. Comparing expression within the isogenic strain panel allows us to identify genes whose expression does not increase linearly as gene copy increases; comparing expression in the panel strains to the closely related euploid reference identifies heritable expression differences in the two strain backgrounds. A comparable panel was developed for laboratory strain W303 aneuploid for Chr 12. As expected, there was little growth difference across the wild-strain panels but a major defect as aneuploidy increased in the W303 laboratory strain (*Figure 5B*).

We measured mRNA and DNA abundance across each panel relative to the paired euploid references and developed a mixture of linear regression (MLR) model to classify genes based on the slope and intercept of the mRNA-gene copy relationships (*Figure 6*, see 'Materials and methods'). Genes in Class 1 show proportionate increases in mRNA abundance as gene copy increases across the strain panel, with a slope of 1.0 and a log$_2$ intercept of 0 that indicates comparable expression in the two euploid strains (*Figure 6A*). These genes therefore show no evidence of dosage compensation or heritably altered expression. Genes in Class 2 also show a linear relationship between mRNA and DNA copy (slope of 1.0) but have an altered intercept that reflects either constitutively reduced (Class 2a, *Figure 6B*) or constitutively elevated (Class 2b, *Figure 6C*) mRNA per gene copy. Thus, Class 2 genes display heritably altered expression but no evidence of dosage compensation. In contrast, genes in Class 3 show a disproportionate relationship between mRNA abundance and gene copy number. For genes in Class 3a, mRNA does not increase proportionately as gene copy increases across the strain panel, as evidenced by reduced slope of the linear fit (*Figure 6D*). Analogously, genes in Class 3b show a slope >1, indicating that mRNA abundance is amplified above expectation as gene copy increases. Because strains within each panel are isogenic aside of the aneuploidy, the reduced slope for Class 3a genes is indicative of dosage compensation while the elevated slope for Class 3b genes represents a disproportionate increase in expression.

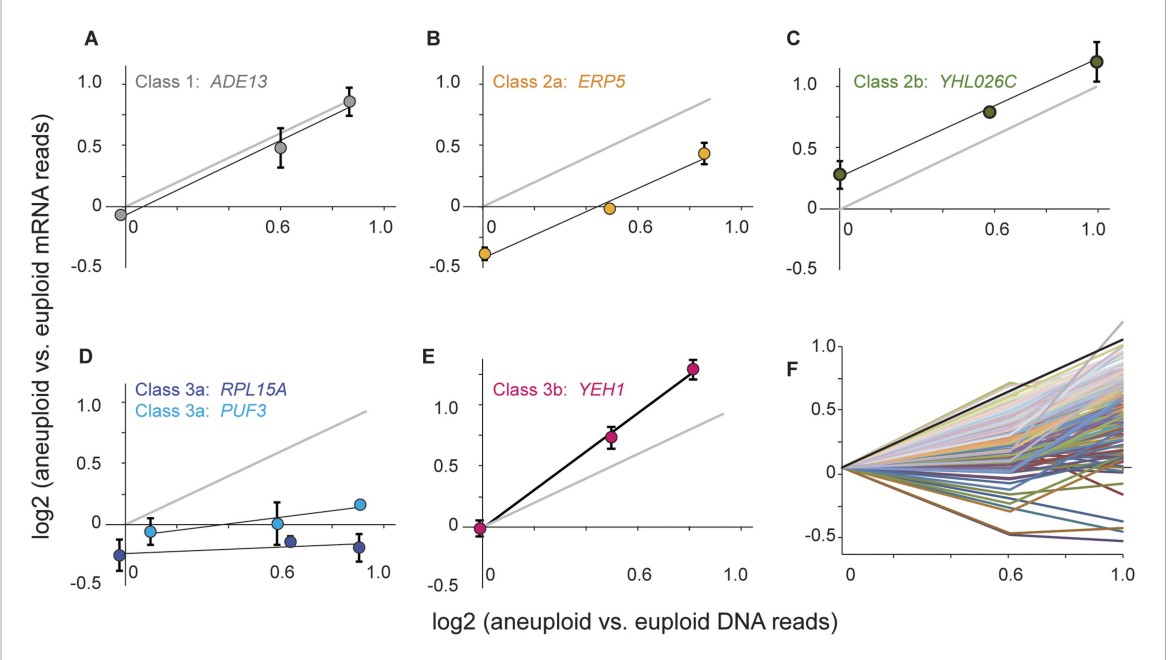

**Figure 6**. Classification of dosage-responsive genes. The $\log_2$(aneuploid vs euploid RNA-seq reads) for each gene was plotted against the $\log_2$(aneuploid vs euploid DNA-seq reads) measured for that gene in the -2n, -3n, and -4n strains within the strain panel normalized to the closely related euploid reference. (**A–E**) Average and standard deviation of representative genes in the denoted classes, across three biological replicates. (**F**) All 173 Class 3a genes, normalized to the euploid strain from within the respective panel for clarity. The expected relationship representing proportionate increases in expression (slope of 1.0, intercept of 0) is shown in each panel.

We applied the MLR model to classify genes based on their maximum posterior probability (see 'Materials and methods'). The results varied to some degree depending on the strain (*Table 1*). In the YPS1009_Chr12 strain panel, 7% of genes showed expected mRNA increases across the strain panel and with regard to the reference strain (Class 1). In contrast, 28% of genes showed heritably reduced expression per gene copy (Class 2a) while only 1% showed heritably increased expression (Class 2b). However, nearly a third of Chr 12 genes displayed a phenotype consistent with dosage compensation (Class 3a). In the NCYC110_Chr 8 strain panel, 11% of genes displayed a phenotype consistent with dosage compensation while almost half the Chr 8 genes were classified as having heritably reduced expression.

**Table 1**. MLR Gene classifications

| Gene class | NCYC110 (Chr 8) | YPS1009 (Chr 12) |
|---|---|---|
| Non-linear | 19 (7%) | 32 (7%) |
| Class 1 | 16 (6%) | 81 (16%) |
| Class 2a | 138 (49%) | 137 (28%) |
| Class 2b | 32 (11%) | 4 (1%) |
| Class 3a | 30 (11%) | 142 (29%) |
| Class 3b | 46 (16%) | 96 (20%) |
| TOTAL | 281 | 492 |

The number and percentage of genes classified in each group. Functional enrichments are listed in **Supplementary file 2**.

## Genes linked to cell morphology show amplified expression in aneuploid strains

Sixteen to 20% of duplicated genes showed amplified mRNA abundance across the strain panels (Class 3b) or in isogenic aneuploid–euploid pairs (*Figure 4B*). As a group, these 206 genes were enriched for genes important for cell morphology (p = 6e-6). We noted that several of our aneuploid strains displayed differences in flocculence or cell shape compared to euploid controls, as previously reported for aneuploid strains (*Wu et al., 2010*; *Tan et al., 2013*). We reasoned that this may be an indirect effect that extends to unamplified genes, and thus we identified unamplified genes with linear expression increase proportionate to the copy number of the amplified chromosome. This identified 39 genes in the YPS1009_Chr12 panel and 96 genes in the NCYC110_Chr8 panel (false discovery rate 0.05, see 'Materials and methods'). When pooled together, and especially when combined with the 206 genes with amplified expression described above, there was strong enrichment for proteins localized to the membrane (p = 5e-4) and for those important for proper morphology (1e-8). Genes encoding cell-surface proteins are known to display altered expression due to changes in cell size/shape induced by ploidy effects (*Wu et al., 2010*), which may explain their altered expression here.

## Dosage compensation occurs at specific functional groups with some variation across strains

To further investigate genes subject to dosage compensation, we combined Class 3a genes identified in the strain panels with genes displaying lower-than-expected expression in the isogenic aneuploid–euploid strain pairs (*Figure 4B*, see 'Materials and methods'), for a total of 245 genes. As a group, these genes were strongly enriched for translation factors (p = 5e-7), cytosolic RPs (p = 1e-5), ribosome biogenesis genes (p = 1e-4), iron/copper transporters (p = 4e-5), guanyl-nucleotide exchange factors (p = 7e-4), and genes encoding mitochondrial proteins (p = 3e-5). Once again, few of these genes (<7%) were repressed in other aneuploid strains, demonstrating that this is not an aneuploidy-stress response.

Three of the isogenic aneuploid-euploid strain groups involved amplification of Chr 8, allowing us to assess strain-specific effects on dosage compensation. Of the genes with lower-than-expected expression in either the haploid YPS163_Chr8-disome or diploid T73_Chr8-4n strain, 79% showed lower-than-expected expression in the NCYC110_Chr8 strain panel. While an over-abundance of these genes were scored as dosage compensated in Class 3a (p = 7e-3), a surprising number of genes called compensated in the paired analysis were classified as having heritably reduced expression without dosage compensation (Class 2a) in the NCYC110_Chr8 panel. One possibility is that the genes are mis-classified or are affected by both heritably reduced expression and a mode of dosage compensation—while the latter was true for a handful of the NCYC110 genes, most displayed high membership in Class 2a and had a slope and fit that was inconsistent with dosage compensation. These results raise the possibility that there exist strain-specific differences in the genes that are subject to dosage compensation (see 'Discussion').

## Dosage compensation is mediated by multiple mechanisms

In the case of several interrogated RP genes, dosage compensation is most likely due to feedback control (*Figure 7*). When cloned onto a low-copy plasmid with flanking intergenic regions, transcripts *RPL15A* and *RPL22A* did not increase despite increasing DNA copy (*Figure 7A,B*). This was true regardless of allele, ploidy or strain background, indicating that the dosage response of RP genes functions in laboratory-strain backgrounds. In contrast, the dosage response was not seen when several mitochondrial genes from Chr 12 were duplicated in isolation, since mRNA for the genes increased ~2× when the genes were amplified (*Figure 7C,D*). To further investigate the mechanism, we deleted the right arm of two of the four copies of acrocentric Chr 12 in the YPS1009_Chr12-4n strain (*Figure 8A*), thereby relieving aneuploidy at ~889 kb and 80% of the Chr 12 genes. For several mitochondrial genes that remained amplified in this strain, the dosage effect was lost, while for other genes expression remained lower than expected (*Figure 8B*). These results suggest a more complicated mechanism that will require further experiments to dissect.

We were unable to fit the MLR model to the W303 panel: the W303_Chr12-4n strain had very abnormal expression, as described above. Surprisingly, however, expression in the W303_Chr12-3n strain was highly correlated to the YPS1009_Chr12-3n strain: for two-thirds of the Class 3a genes,

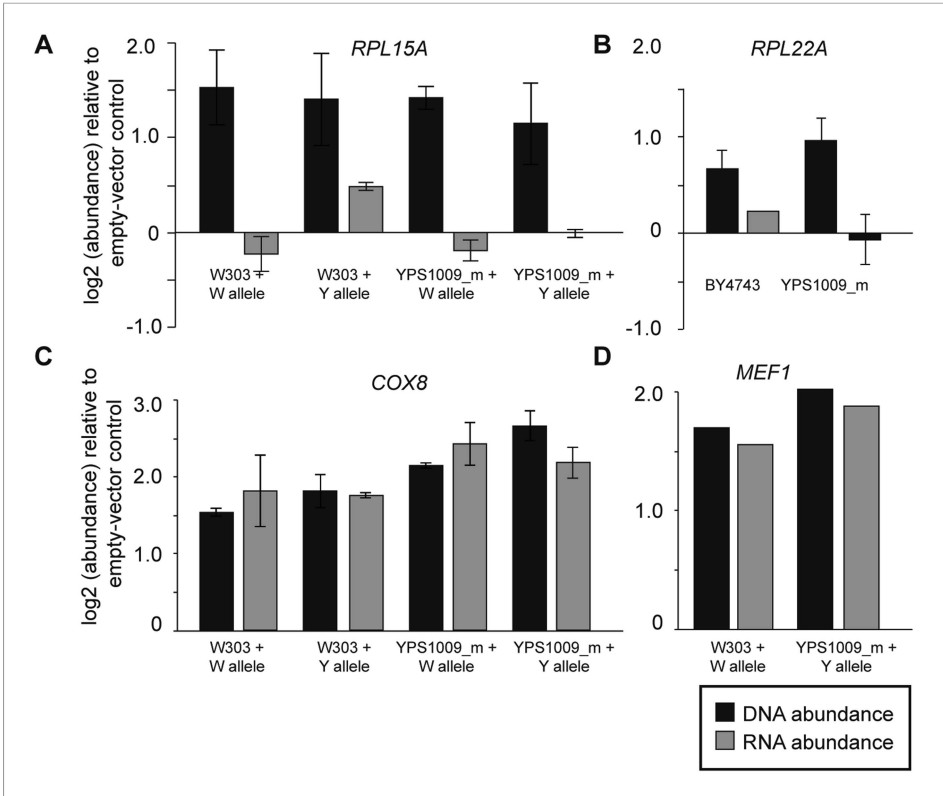

**Figure 7**. Expression response when genes are duplicated on a plasmid. To measure the effects of gene duplication in the absence of aneuploidy, representative Class 3a genes were cloned onto a CEN plasmid and introduced into otherwise euploid strains. As a control, strains were also compared to an empty vector for normalization. mRNA and DNA abundance for the gene were quantified by qPCR as described in 'Materials and methods'. Relative DNA (black) and RNA (gray) abundance for (**A**) *RPL15A*, (**B**) *RPL22A*, (**C**) *COX8*, or (**D**) *MEF1* when introduced in extra copy into each strain. Panels **A**, **C**, **D** show abundance in haploid W303 or the haploid, monosomic (euploid) derivative of YPS1009 ('YPS1009_m'), and panel **B** shows abundance in diploid laboratory strain BY4743 and diploid, euploid derivative YPS1009_Chr12-2n. In all cases, the average and standard deviation of biological triplicates is shown.

expression in W303_Chr12-3n was within 10% of YPS1009_Chr12-3n values. Thus the dosage compensation phenotype appeared in effect at some genes in the W303_Chr12-3n strain but was lost or obscured in the W303_Chr12-4n strain. One clear exception was the group of eleven mitochondrial genes on Chr 12 that showed lower-than-expected expression in the YPS1009_Chr12 panel but proportionately higher (or amplified) expression in the W303_Chr12 panel (*Figure 8C*). These results once again point to a difference in mitochondrial function in W303 compared to other strains.

## Genes subject to dosage compensation are buffered against CNV

We sought to identify common features of dosage-compensated genes that may explain their tighter expression control. One prediction is that dosage compensation occurs at genes that are most toxic when overexpressed. Indeed, the combined set of 245 dosage-compensated genes is enriched for genes that are deleterious in very high copy in the laboratory strain (p = 0.009, *Makanae et al., 2013*). A second prediction is posited by the balance hypothesis (*Birchler and Veitia, 2007*), which asserts that expression of multi-subunit protein complexes may be more tightly controlled to maintain protein stoichiometry. We found weak enrichment for proteins in multi-subunit complexes (p = 0.03, *Pu et al., 2009*), but the significance eroded if RPs were removed from the analysis. The group of dosage-compensated genes displayed slightly higher transcript abundance and higher RNA polymerase occupancy than the average gene (*Chasman et al., 2014*), but the trends did not hold if genes related to translation were removed.

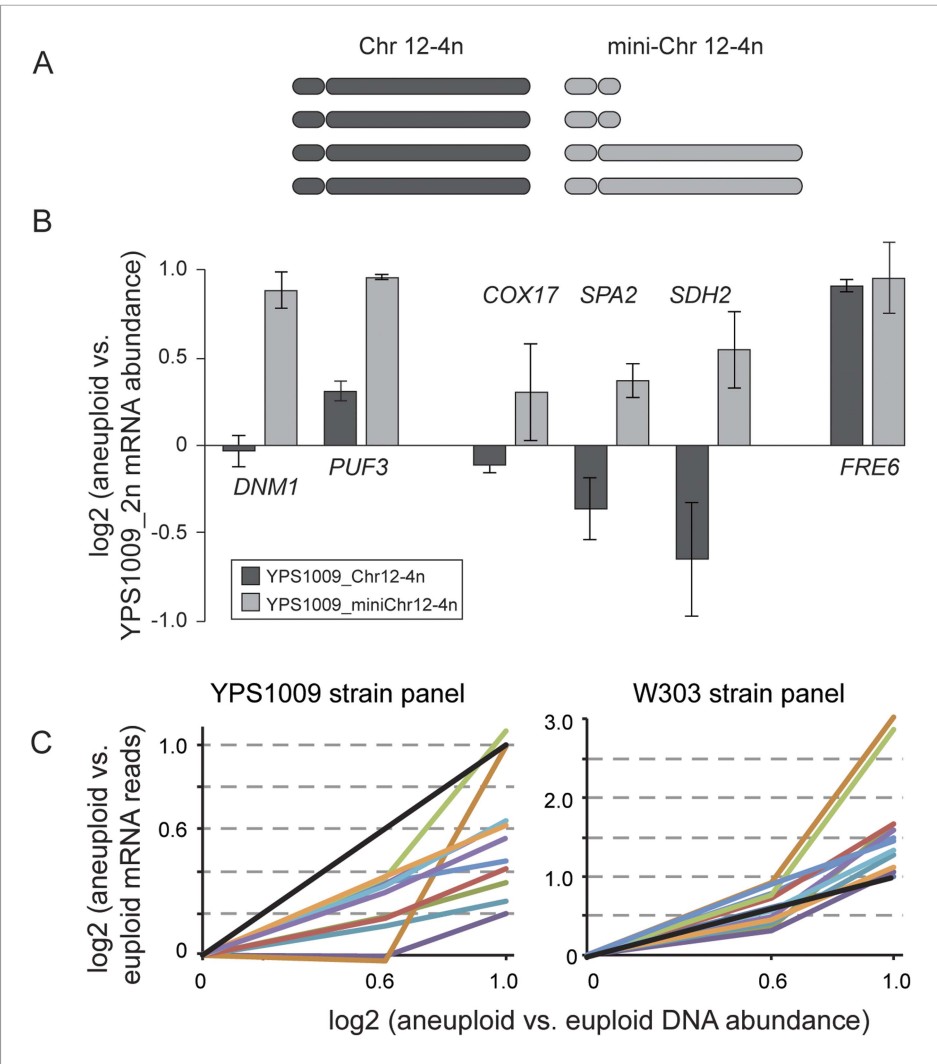

**Figure 8**. Relative mRNA abundance of amplified genes in the mini-4n strain and strain panels. (**A**) Cartoon diagram of the four copies of Chr 12 in the tetraploid YPS1009_Chr12-4n strain and in the 'mini_4n' strain where two copies of the Chr 12-right arm are deleted (see 'Materials and methods' for details). (**B**) Expression was measured on tiled yeast-genome DNA microarrays. Shown are Class 3a genes that remain amplified in the mini-4n strain and could be quantified by arrays. Relative mRNA abundance was measured in biological triplicate in the YPS1009_Chr12-4n or YPS1009_mini-Chr12-4n strain vs the euploid YPS1009_Chr12-2n strain. Genes *COX17*, *SPA2*, and *SDH2* showed an increase in expression in the mini_4n strain, however expression remained significantly lower than the expected two-fold difference proportionate to the gene amplification. *FRE6* showed little dosage compensation when measured by DNA microarray analysis and correspondingly its expression did not change in the mini_4n strain. (**C**) Relative mRNA abundance of eleven genes that are dosage compensated across the YPS1009_Chr12 strain panel (left) but not the W303_Chr12 strain panel (right).

Dosage-compensated genes were enriched for genes that are toxic when highly abundant, raising the possibility that their expression may also be under greater evolutionary constraint. To investigate this, we compared the variance in gene expression seen in natural isolates subject to mutation and selection (the genetic variance, $V_g$) (*Skelly et al., 2013*) to mutational variance ($V_m$) from MA lines (*Landry et al., 2007*). Genes that are under the highest constraint will have negative $\log_2(V_g/V_m)$ ratios reflecting that expression variation is being removed by purifying selection. Genes subject to dosage compensation and genes affected by heritably reduced expression show higher constraint compared to all genes, as expected for genes that are toxic when over-expressed (*Figure 9A*). The effect was also true for amplified genes that display proportionate expression upon amplification, which is likely

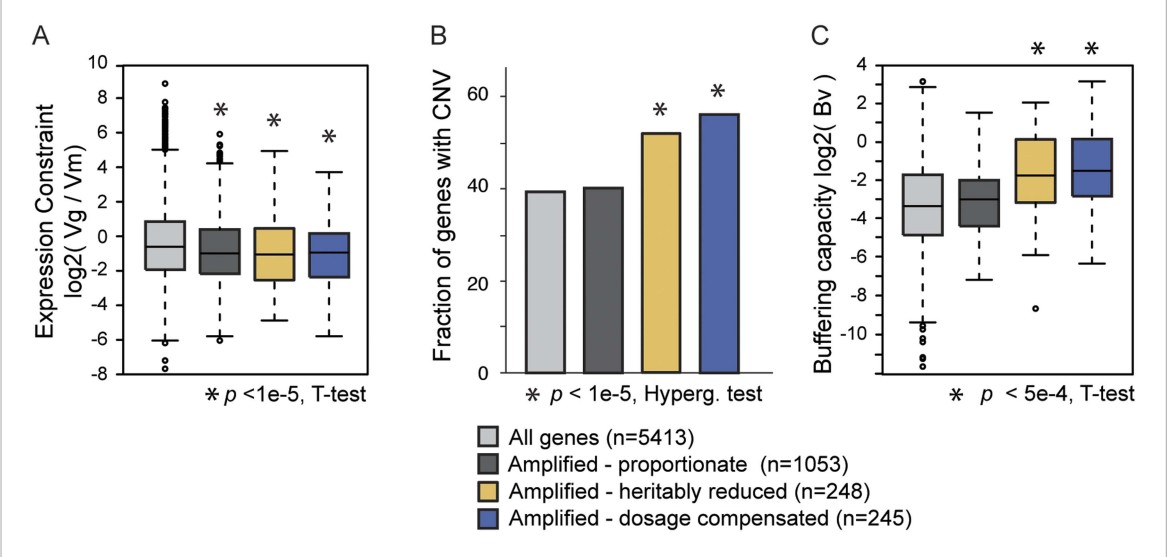

**Figure 9**. Natural variation in expression constraint and gene copy-number variation. (**A**) $\log_2(V_g/V_m)$ as described in the text, (**B**) fraction of genes with copy-number variation (CNV) in at least one of 103 strains, and (**C**) log2 of the buffering capacity score, $Bv$, are shown for the gene groups in the key (where number of genes is indicated in parentheses). Amplified genes with proportionately higher expression include Class 1 genes and genes with proportionate expression in strain pairs (*Figure 4A*, gray points). Amplified genes with heritably reduced expression correspond to genes in Class 2a, minus genes identified as dosage compensated in other strains, while amplified genes subject to dosage compensation represent genes in Class 3a plus genes with lower-than-expected expression as identified in *Figure 4B*. Statistical significance was scored by comparing each group to the total set of yeast genes.

a byproduct of our classification (since we effectively eliminated genes with variable expression across strains and replicates). The trends were consistent when genes belonging to all enriched functional groups were removed from the analysis.

An active mode of dosage compensation is predicted to constrain expression in the face of underlying gene amplification, thereby buffering CNV. To test this hypothesis, we cataloged gene amplifications measured by array-CGH in 103 strains (*Dunn et al., 2005*; *Carreto et al., 2008*; *Dunn et al., 2012*). In fact, dosage-compensated genes show considerably higher CNV than genes that display proportionate mRNA increase upon gene amplification (p = 4e-8, *Figure 9B*). Somewhat surprisingly, genes with heritably reduced expression also showed relaxed constraint in CNV, which may reflect that some of these genes are dosage compensated in other strain backgrounds (see 'Discussion'). Once again, the trends were not driven by enriched functional groups. We devised a CNV-buffering score, $B_v$, for each gene as the phylogeny-weighted sum of gene copy number across strains divided by $V_g/V_m$ measured for that gene (see 'Materials and methods')—genes with the strongest buffering capacity will therefore have the largest scores. Genes subject to dosage compensation show significantly higher $B_v$ scores compared to all genes and compared to amplified genes with proportionately higher expression (p = 1.7e-4). Amplified genes with heritably reduced expression also showed higher buffering scores albeit lower than dosage-compensated genes (see 'Discussion'). Together, these results show that genes subject to dosage-compensated expression can be buffered against CNV in *S. cerevisiae* populations.

## Chr 12 amplification provides a selective advantage for an ecological trait

The ability to buffer the toxicity of gene amplification could facilitate rapid evolution through aneuploidy. Aneuploidy in laboratory strains can be advantageous under adverse conditions, when non-dosage compensated defense genes are amplified in expression (*Hughes et al., 2000*; *Dunham et al., 2002*; *Chen et al., 2012*; *Yona et al., 2012*); but the extent to which this occurs in nature is not known. We and others have previously shown that strains of *S. cerevisiae* and *Saccharomyces paradoxus* from wintry climates have undergone selection to maintain freeze-thaw (FT) tolerance

(*Will et al., 2010*; *Leducq et al., 2014*). We noted that the *AQY2* water-transporter gene underlying FT tolerance (*Will et al., 2010*) resides on Chr 12, which is amplified in New Jersey-oak strain YPS1009_Chr12-3n. *AQY2* escapes dosage compensation, resulting in higher *AQY2* expression across the strain panel (*Figure 10A*). Indeed, we found that FT resistance improves with increasing Chr 12 copy number in YPS1009 (*Figure 10B*). The strong signatures of selection that we previously observed at the *AQY2* gene (*Will et al., 2010*) and the clear relevance of the trait suggest that Chr 12 amplification is advantageous to YPS1009 in nature.

## Discussion

Our results provide new insight into a long-standing conundrum: how do cells tolerate duplication of toxic genes long enough for new phenotypes to evolve? Our results show that up to 40% of amplified genes in naturally aneuploid strains show lower-than-expected expression. The reduced expression is in part due to heritable polymorphisms that down-regulate expression from amplified genes, some of which may have been selected for during the adaptation to aneuploidy. But reduced expression of other amplified genes—up to 30–36% in some strains—appears to be actively regulated in proportion to increased gene dosage. The prevalence of the dosage compensation found here is likely true for other species and may explain the lack of expression increase previously noted for amplified autosomal genes in *Drosophila* (*Zhang et al., 2010*; *Zhou et al., 2011*) and humans (*Antonarakis et al., 2004*; *Schuster-Bockler et al., 2010*; *Wang et al., 2011*; *Woodwark and Bateman, 2011*).

The clear enrichment of specific functional classes—including genes encoding RP and mitochondrial proteins—points to specific targeted processes. While we did not find overwhelming support for the balance hypothesis, dosage compensated genes include several multi-subunit protein complexes and pathways. In the case of RP genes, dosage compensation most likely occurs via feedback to modulate mRNA abundance (*Figure 7*). Feedback is known to occur for several RP genes, including excess, unassembled L32 that binds a stem loop structure in its own transcript and inhibits intron splicing, perhaps triggering mRNA degradation (*Dabeva and Warner, 1987*; *Vilardell and Warner, 1997*). Our results imply that feedback pertains more broadly to other RP genes, including *RPL15A* (*Figure 7A*)—notably, this transcript lacks an intron and thus must utilize a mechanism distinct from *RPL32*. The mechanism of dosage compensation at nuclear-encoded mitochondrial genes is less clear; however, several mitochondrial proteins are known to regulate translation or stability of other transcripts in shared pathways. For example, unassembled Cox1 protein can suppress translation of the *COX1* transcript (*Van Loon et al., 1983*; *Mick et al., 2011*), while several TCA-cycle enzymes double as RNA-binding proteins that report on TCA activity (*Elzinga et al., 1993*; *de Jong et al., 2000*). A mode of dosage compensation that functions at the level of transcript abundance may provide an additional level of regulation to control protein stoichiometry.

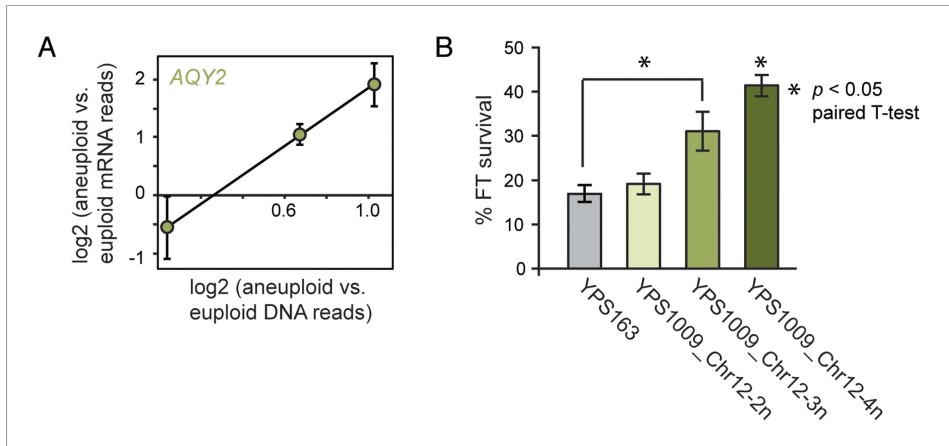

**Figure 10**. Amplified expression of *AQY2* provides a fitness advantage. (**A**) *AQY2* expression as described in *Figure 6*. (**B**) Average and standard deviation of percent cell viability after freeze-thaw stress.

Natural isolates are generally tolerant to aneuploidy, whether or not they have had time to adapt to the extra chromosome. We did observe a common expression response in different aneuploid strains, including the up-regulation of oxidoreductases and the down-regulation of genes involved in mitochondrial translation and respiration. Interestingly, these trends are also seen in DS cells, which are extremely sensitive to reactive-oxygen species (ROS), and perhaps consequently, down-regulate mitochondrial respiration that in turn limits ROS production (*Helguera et al., 2013*). We found no defect in respiratory capacity in naturally aneuploid yeast, and so the implications of this response in yeast are not clear. Human patients display significant variability in their sensitivity to DS, suggesting genetic effects on DS pathologies (*Roper and Reeves, 2006*; *Prandini et al., 2007*). Our results also implicate strain-specific responses to Chr 8 amplification, in particular in the genes subject to dosage compensation vs heritably reduced expression. While some of the Class 2a genes could be mis-classified, these results raise the possibility of natural genetic variation in dosage compensation.

While the expression of dosage-compensated genes is controlled in proportion to gene copy, many genes escape dosage compensation in response to aneuploidy and can contribute significantly to phenotypic variation. Whole and partial chromosome amplification is common in laboratory selection experiments (*Dunham et al., 2002*; *Gresham et al., 2008*; *Yona et al., 2012*) and is particularly prevalent in laboratory mutants that have an extreme fitness defect (*Hughes et al., 2000*; *Koszul et al., 2004*; *Rancati et al., 2008*). Yet the *Saccharomycotina* karyotype is extremely stable, holding at eight or sixteen chromosomes in most pre- and post-genome duplication species, respectively. This suggests that aneuploidy may serve as a transient intermediate, one that can be readily generated—and readily lost—depending on the selective pressures. *Yona et al. (2012)* showed that selection for increased stress tolerance in the laboratory produced aneuploid strains that eventually gave way to other solutions even when the selective pressure persisted. Our results show that aneuploidy is frequent and tolerated in nature and may provide an important route toward natural genetic variation.

## Materials and methods

### Strains and growth conditions

*S. cerevisiae* strains listed in *Supplementary file 1* (diploid unless otherwise noted) were grown at 30°C in batch culture in YPD (1% yeast extract, 2% peptone and 2% dextrose) medium. We chose this growth regime as opposed to chemostat cultures because our strains show little growth differences across the strain panels and also to avoid confounding effects in interpreting gene expression. Given the instability of chromosomal amplification in W303, aneuploid W303 strains were grown under appropriate selection to maintain the aneuploidy and shifted to YPD batch culture growth at 30°C depending on the experiment (outlined below). Growth rates shown in *Figure 1B* were obtained on a Tecan Infinite M200 PRO instrument (Tecan Mannedorf, Switzerland), growing cells in 96-well plates without shaking and scoring doubling times using the program GrowthRates (*Hall et al., 2014*). Growth rates shown in *Figure 1B* for each aneuploid strain were normalized to the average doubling time for euploid strains in the same ecological group (e.g., clinical, natural, oak, vineyard, industrial, sake groups). Doubling times shown in *Figures 1C, 3, and 5* were determined from batch cultures.

### Forcing aneuploidy in euploid parents

We followed the protocol outlined by *Chen et al. (2012)* to isolate aneuploid derivatives. Haploid strains (derived from S288c (DBY7286), oak strain YPS163 or vineyard strain KCY40) were grown in YPD +20 µg/ml radicicol (AG Scientific, San Diego, CA) for 24 hr to promote chromosome instability and induction of aneuploidy. Cells were then washed 3X with YPD, grown in YPD for 24 hr, and plated on YPD + 8–32 µg/ml fluconazole to select for Chr 8 aneuploidy, since amplification of *ERG11* on Chr 8 was shown to confer fluconazole resistance (*Chen et al., 2012*). Minimal inhibitory concentrations on fluconazole at which 90% of cells (MIC90) die were determined for each strain: 32, 8, and 16 µg/ml for DBY7286, YPS163 and KCY40, respectively. Cells with Chr 8 amplification were initially screened via qPCR of select Chr 8 genes and confirmed by array comparative genomic hybridization (aCGH).

### Generation of Aneuploid isogenic strain panels

We generated isogenic aneuploid-euploid strain pairs from two strains, T73_Chr8-4n and YJM428_Chr16-4n (derived from a spore of the original parent, YJM428_Chr12-3n_Chr16-4n in which Chr 12 amplification was lost). T73_Chr8-4n was passaged in liquid YPD culture for

~275 generations before a euploid derivative was identified by qPCR screening. YJM428_Chr16-4n was passaged for ~160 generations to produce a triploid strain that was then dissected and underwent mating-type switching to generate a diploid, euploid YJM428_Chr16-2n derivative.

We also generated two wild-strain panels from YPS1009 (also referred to as YPS1009_Chr12-3n for clarity in the text) and NCYC110 (aka NCYC110_Chr8-4n), and one laboratory strain panel for diploid W303_Chr12-3n (generated by mating a haploid disomic strain, generously provided by A Amon, to the euploid W303). Strain panels for W303_Chr12 and YPS1009_Chr12 were generated by sporulating the trisomic parents and selecting haploid spores with either one or no extra copies of Chr 12. The YPS1009 spores then diploidized after mating-type switching to form isogenic diploid strains having two (2n, i.e., euploid) or four (4n) copies Chr 12. The haploid W303_Chr12 spores were crossed to appropriate W303 strains to generate isogenic diploids with two, three, or four copies of Chr 12. Homozygous strain NCYC110_Chr8-4n was passaged in liquid YPD for ~427 generations before an isolate with only three Chr 8 copies was identified by qPCR screening. (A similar procedure was performed for YPS1009_Chr12-4n and W303_Chr12-4n to determine the stability of the aneuploidies.) A completely euploid derivative of NCYC110 was isolated shortly after at 444 generations. Aneuploidy of the respective chromosomes in each of the strain panels was initially verified via aCGH and subsequently via genomic sequencing, with periodic confirmation by qPCR to ensure aneuploidy maintenance.

## Genomic sequencing

Genomic sequencing was performed on genomic DNA isolated with a Genomic-tip 20/G Kit (Qiagen, Germantown, MD). 1 µg genomic DNA in a total volume of 50 µl in a 0.5 ml microcentrifuge tube was fragmented with a Diagenode Bioruptor sonication device (Diagenode, Denville, NJ) to a peak fragment size of 300–400 bp, and 1 µg was used as an input into Illumina's TruSeq DNA Library Prep (Illumina, San Diego, CA). Ligation products were purified with the E-Gel SizeSelect System (Life Technologies, Carlsbad, CA). All cleanup steps in the genomic library prep were performed with Axy Prep MAG PCR Cleanup beads (Corning, Corning, NY). Genomic libraries were sequenced on Illumina's HiSeq 2000 or HiSeq 2500 System, generating single end 100 bp reads. Sequencing reads were processed with Trimmomatic version 0.30 (*Bolger et al., 2014*), and reads were mapped to a reference genome with strain-specific SNPs (for YPS163, YPS1009, NCYC110, NCYC3290, and W303) or to the S288c reference 58 for all other strains, using BWA version 0.7.3 (*Li and Durbin, 2009*). HTseq version 0.5.4 (*Anders et al., 2014*) summed read counts per gene, which were then normalized for gene size and the number of reads generated per library via reads per kilobase per million-mapped reads (RPKM). Genomic DNA was sequenced in duplicates for all strain panels. Aneuploid strains were identified as triploid or tetraploid if the median RPKM signal across the chromosome was 0.4–0.7 (to call triploids) or 0.7 to >1.0 (to call tetraploids). Genome sequences are available in the NIH Sequence Read Archive (SRA) under accession SRP047341.

To identify potential polymorphisms in nominally isogenic aneuploid–euploid pairs, SNPs in each parental strain were called with GATK (*DePristo et al., 2011*) and substituted into the S288c genome reference, which was then used as the reference for remapping of all related strains. GATK was used to call single-nucleotide polymorphisms (SNPs) in each remapped strain. We found one homozygous SNP in NCYC110_Chr8-3n and a different homozygous SNP in NCYC110_Chr8-2n that were each identified in both replicates of the strain sequences. We performed a similar procedure for singleton genome sequences from YJM428_Chr16-2n and T73_Chr8-2n and identified 13 or 40 homozygous SNPs; these numbers are in the range of false-positive identifications found in a single replicate, as assessed from the NCYC110_Chr8 analysis above. We conclude from this analysis that there are relatively few legitimate SNPs in passaged strains and that the majority of expression differences are a direct response to the aneuploidy.

## RNA-seq

Cells were harvested for 3 min at 3000 r.p.m., after which time the cells were flash frozen in liquid nitrogen and maintained at −80°C until RNA extraction. Select samples (for T73_Chr8, YPS163_Chr8, and YJM428_Chr16 strain pairs and one replicate of the NCYC110_Chr8 strain panel) were collected and mixed at a 10:1 cellular ratio with *Schizosaccharomyces pombe* PR100, which had been grown in YES medium for >7 generations to $OD_{600}$ ~0.5, killed with 0.125V ice-cold quench solution (5% acid phenol in 100% EtOH), and collected. Cells were carefully counted on a hemocytometer to estimate

cell counts. Total RNA was extracted with hot phenol as previously described (*Gasch, 2002*), DNase-treated at 37°C for 30 min with TURBO DNase (Life Technologies), and then precipitated at −20°C in 2.5 M LiCl for 30 min rRNA depletion of the DNase-treated total RNA and subsequent cDNA library preparation were performed with ScriptSeq Complete Kit H/M/R (Epicentre, Madison, WI), Index PCR Primers (Epicentre) and FailSafe PCR Enzyme Mix (Epicentre). rRNA-depleted RNA was purified with a RNeasy MinElute Cleanup Kit (Qiagen), while cDNA was purified with Axy Prep MAG PCR Cleanup beads (Corning). cDNA libraries were sequenced on Illumina's HiSeq 2000 System (UW-Madison DNA Sequencing Facility), generating single-end 100 bp reads. Sequencing reads were processed with Trimmomatic version 0.30 (*Bolger et al., 2014*) and reads were mapped to a reference genome with strain-specific SNPs (for YPS163, YPS1009, NCYC110, NCYC3290, and W303), to the S288c reference concatenated with the *Sz. pombe* genome (for doped samples listed above), or to the S288c reference 58 for all other strains, using BWA version 0.7.3 (*Li and Durbin, 2009*). HTseq version 0.5.4 (*Anders et al., 2014*) was used to obtain read counts per gene. Sequencing was done in biological triplicate for strain panels or biological duplicate for all strains except sake strains, which were done as singlets, with paired growth and strain collection for each replicate. For samples spiked with *Sz. pombe* controls, reads were normalized by a scaling factor such that the slope of the *Sz. pombe* reads across samples was 1.0, or normalized by traditional RPKM. The normalization methods produced data that were virtually indistinguishable; because RPKM-normalized data agreed better across biological replicates, RPKM normalization was used for all analyses described here. All RNA-seq data are available at NIH GEO accession GSE61532.

Transcriptome profiling was done for six aneuploid strains and paired controls (including YJM428 and YJM308, Y2189 and Y2209, K9 or K1 and K10, NCYC110 and NYCY3290, and YPS1009 and YPS163). In the case of K9/K1, NCYC110, and YPS1009, the control was chosen based on the genetically closest known relative at the time of the analysis based on phylogenetic comparisons (*Fay and Benavides, 2005*; *Kvitek et al., 2008*; *Liti et al., 2009*). Controls for clinical isolate YJM428 and natural strain Y2189 were based on ecological group, choosing another clinical or natural isolate, respectively. RPKM values from each aneuploid were divided by RPKM measured from the euploid controls and logged, for both mRNA and DNA samples. For the analysis shown in *Figure 4*, we randomly chose two of the three replicates of YPS1009_Chr12-3n and NCYC110_Chr8-4n to produce statistical power comparable to the other strains analyzed in duplicate. Genes with lower-than-expected expression per gene copy (*Figure 4*) were identified as follows:

For each aneuploid chromosome in each strain, we first removed sub-telomeric genes with skewed measurements since these genes frequently show copy-number differences across strains. We then calculated the chromosome-wide mean and standard deviation of the $\log_2$(aneuploid vs euploid DNA reads), across all genes on the affected chromosome. For each amplified gene being considered, we took the mean of the $\log_2$(aneuploid vs euploid DNA reads) measured for that gene specifically; we used this value minus one standard deviation of the chromosome-wide mean (or two standard deviations in the case of the sake strains)—this value served as a gene-specific cutoff for relative mRNA abundance. Genes with lower-than-expected expression per gene copy were identified if the $\log_2$(aneuploid vs euploid mRNA reads) was less than the filtering cutoff in both replicates (or one replicate for sake strains, which used a more stringent cutoff). This process identified genes whose relative mRNA abundance was lower than the relative DNA abundance at a high confidence interval. Genes whose $\log_2$(aneuploid vs euploid mRNA reads) difference was more than 1.5X lower (0.6 in $\log_2$ space) than the euploid control were excluded from the list, since their expression may be influenced by other effects. This identified 838 unique genes from the six aneuploid strains whose expression was lower-than-expected per gene copy, with 111 genes amplified and affected in >1 strain. We also identified amplified genes whose expression was distinctly not affected by dosage compensation as genes whose relative $\log_2$(mRNA abundance) was within the filtering cutoff for both replicates, excluding genes whose $\log_2$ expression difference was more than 2.5× higher than the euploid control, since their expression may be influenced by other effects. This identified 928 genes whose expression was in the expected range if *no* dosage compensation is at work. A subset of these genes showed expression that was at least 1.5× higher than expected, and these are annotated as magenta points in *Figure 4*.

Genes that were differentially expressed in aneuploid vs euploid strains (regardless of DNA abundance) were identified with the program edgeR (*Robinson et al., 2000*). Because of limited power for the duplicated data sets, we also included genes whose expression was greater than

1.3X ($\log_2$ of 0.4) different from the euploid control in three of the six strains. Clustering of expression was done using the program Cluster 3.0 (http://bonsai.hgc.jp/~mdehoon/software/cluster/software.htm) and visualized by Java Treeview (http://jtreeview.sourceforge.net). Enrichment of functional groups was done using the program FunSpec (*Robinson et al., 2002*). Enrichment of proteins in complexes was scored by hypergeometric test, comparing to curated complexes from (*Pu et al., 2009*); differences in degree from protein interaction networks was scored by T-test comparing the degree compiled by (*Chasman et al., 2014*) from genes in Class 3a vs other gene groups.

## Mixture of linear regressions model

A novel MLR model was developed to classify genes based on their expression across the two wild-strain panels. Preprocessed mRNA and DNA abundance data were analyzed further in the context of a new mixture of linear regressions (MLR) model in order to cluster genes according to their DNA/mRNA relationship. Within a given strain panel we modeled the relationship between DNA abundance at gene g from sample i, say $X_{g,i}$, and mRNA abundance $Y_{g,i}$, where both are normalized to euploid controls and considered on the logarithmic scale. Briefly, we first filtered genes that exhibited a nonlinear relationship using a likelihood ratio test (*Supplementary file 3*). The interpretation of these profiles is not clear and therefore they were removed from further consideration. Data from the remaining genes, which exhibited a linear relationship $Y_{g,i} = A_g + B_g X_{g,i} + error_{g,i}$, were fit to a discrete, random-effects mixture model in order to produce a relationship classifier (*McLachlan and Peel, 2000*). In the proposed MLR model, the five discrete classes correspond to constraints on the intercept $\alpha$ and slope $\beta$ parameters of the linear regression: (Class 1) $\alpha_g = 0$, $\beta_g = 1$, (Class 2a) $\alpha_g < 0$, $\beta_g = 1$, (Class 2b) $\alpha_g > 0$, $\beta = 1$, (Class 3a) $\beta < 1$, (Class 3b) $\beta_g > 1$. Class 1 represents genes with no dosage compensation and statistically indistinguishable expression between the 2n strain and paired euploid control. Class 2 represents genes whose expression increases linearly across the strain panel, but whose expression in the aneuploid derivatives is lower (Class 2a) or higher (Class 2b) per gene copy than the paired euploid. In contrast, Class 3 genes show expression changes across the strain panel that are disproportionate to the change in DNA copy, either showing lower (Class 3a) or greater (Class 3b) expression per gene copy as the aneuploidy increases. The inference is stabilized by treating the gene-specific slopes and intercepts as random effects that are constrained by the class structure; this effectively reduces the dimension of the parameter space. Fortuitously, explicit formula is available for the probability density of data within each class. The expectation maximization (EM) algorithm was used to estimate the MLR parameters; subsequently, genes were clustered together if they had a high posterior probability of arising from the same discrete class. Genes were classified based on the maximum posterior probability (*Supplementary file 3*). Further details are provided in Appendix 1. Unamplified genes in the two strain panels that displayed linear increases in expression in proportion to the amplified chromosome copy number were identified based on the probability of linear fit in R, using Benjamini and Hochberg multiple test correction (*Benjamini and Hochberg, 1995*).

## Gene cloning, Chr 12 arm deletion, and qPCR

W303 and/or YPS1009 alleles of each query gene were cloned onto a KANMX-marked CEN plasmid via homologous recombination in strain BY4741, from which plasmids were isolated and used to transform haploid or diploid versions of YPS1009_Chr12-2n, euploid W303, or euploid BY4743, where noted. Strains were grown in YPD + G418 for at least 3 generations to OD$_{600}$ ~0.3. Cells were collected for genomic-DNA and RNA preps to measure copy number and transcript levels, respectively, as described above. RNA was DNase-digested and subsequently used to synthesize cDNA with an Oligo (dT) primer and Superscript III reverse transcriptase (Invitrogen). Quantitative PCR (qPCR) was performed with SYBR-Green as previously described (*Lee et al., 2011*). mRNA or DNA abundance measured for each gene was normalized to an internal control (*ERV25*) and compared to a strain carrying an empty vector.

To test the effect of aneuploidy dose on gene expression, a strain carrying an extra copy of the left-arm of Chr 12 only was generated (referred to a the 'mini-4n strain'), by deleting the right arm of Chr 12 from the YPS1009_Chr12-4n strain, as follows: the KANMX cassette was PCR amplified with homology to the promoter region of *GAT3*, which flanks the Chr 12 centromere, or the promoter region of subtelomerically encoded gene YLR460C. The diploid, aneuploid YPS1009_Chr12-4n strain was transformed and G418-resistant colonies were selected and sporulated. One spore showed the expected 2/4 or 0/4 spore survival when dissected onto YPD + G418. PCR was used to confirm

integration of the KANMX cassette and aCGH was used to confirm that the diploid cells are aneuploid for the left arm of Chr 12, but not the right arm. Expression levels of the amplified Class 3a genes were assessed by qPCR or DNA-microarray analysis as described above.

## Expression constraint and CNV in wild strains

$V_g/V_m$ values were calculated for each gene as follows: genetic variance $V_g$ was taken as the variance in expression across 22 wild strains analyzed in (*Skelly et al., 2013*). $V_m$ was taken as the variance in expression across MA lines generated in the absence of selection by (*Landry et al., 2007*). The ratio of $V_g/V_m$ represents the effects of mutation + selection as measured in natural strains vs the effects of mutation in the near absence of selection as measured in MA lines. CNV data were taken from (*Dunn et al., 2005*; *Carreto et al., 2008*; *Dunn et al., 2012*), which measured gene copy number by aCGH. Strains whose aCGH profiles were correlated >0.9 were represented by a single, randomly chosen strain, to avoid over-representing strains that are highly related. This left 103 strains with aCGH values for each gene. Amplified genes were identified if the relative gene abundance was at least 1.6× higher than the S288c reference. Genes that showed CNV in at least one of the 103 strains were compared in *Figure 9B*. To calculate $B_v$, a measure of the buffering capacity of each gene, we summed the number of strains in which the gene was amplified, weighted by the aCGH similarity weights (described in http://bonsai.hgc.jp/~mdehoon/software/cluster/software.htm) as a proxy for strain relatedness. The weighting was applied to control for genetic similarities in the strains, to therefore highlight independent gene amplification events. For each gene, this value was divided by its calculated $V_g/V_m$ ratio. Larger values represent a higher propensity for CNV coupled with more stringent expression constraint (i.e., small $V_g/V_m$).

## FT tolerance

The YPS1009 isogenic strains and YPS163 were grown in YPD to $OD_{600}$ ~0.3, and 100 μl of cells was subjected to dry ice/ethanol bath freezing (<−50°C) or ice as previously described (*Will et al., 2010*). Viability was determined using Live/Dead straining (Life Technologies) on a Guava EasyCyte Plus flow cytometer (Guava Technologies, Inc., Hayward, CA). Percentages of cells surviving freeze/thaw are reported for 3 biological replicates.

## Acknowledgements

We thank A. Amon for providing the haploid W303 aneuploid strain, M. Place for informatics support, and D. Gottschling and members of the Gasch Lab for useful discussions. This work was supported by NIH grants R01GM083989 (to APG) and R21HG006568 (to MAN), with partial support from the Department of Energy Great Lakes Bioenergy Research Center (DE–FC02–07ER64494) and fellowships from Morgridge Institute of Research for ZW and NSF for M. Sardi.

## Additional information

### Funding

| Funder | Grant reference | Author |
| --- | --- | --- |
| National Institutes of Health (NIH) | R01GM083989 | James Hose, Chris Mun Yong, Audrey P Gasch |
| U.S. Department of Energy (Department of Energy) | DE-FC02-07ER64494 | Maria Sardi, Audrey P Gasch |
| National Institutes of Health (NIH) | R21HG006568 | Zhishi Wang, Michael A Newton |

The funders had no role in study design, data collection and interpretation, or the decision to submit the work for publication.

### Author contributions

JH, Conceived of the experiments, Generated strain panels, conducted RNA and DNA sequence and initial analysis; CMY, Generated the mini-4n strain, Assisted in mini-4n analysis, Assisted in cloning and qPCR quantification; MS, Conducted genome sequencing and growth analysis; ZW, MAN, Conceived of and implemented statistical models and analysis; APG, Conceived of the experiments, Performed all downstream analyses, Wrote the manuscript with input from all authors

## Additional files

### Supplementary files

• Supplementary file 1. Strains used in this study.

• Supplementary file 2. Functional enrichments and p-values for gene classes. All enrichments with p < 2e-3 are shown; enrichments that meet the Bonferroni cutoff (2e-5) are highlighted in red. Enrichments are listed for (a) amplified genes with expression proportionate to gene copy (from *Figure 4A*), (b) pooled set of amplified genes with expression proportionate to gene copy (from *Figure 9*), (c) dosage compensated genes (from *Figure 9*). (d) Genes in Class 2a, (e) genes with higher-than-expected expression (from *Figure 4A,B* plus Class 3b genes).

• Supplementary file 3. Genes in each class are listed on file tabs. The first two columns list gene name and annotation, while the third column lists the strain and/or analysis that the gene was identified in. (a) Amplified genes with expression proportionate to gene copy. (b) Amplified genes with lower-than-expected expression. (c) Amplified genes with higher-than-expected expression. (d) Genes in each MLR classification group—intercept ('alpha-hat'), slope ('beta-hat'), and posterior probabilities for each class are shown. The mean DNA and mRNA sequencing reads for each gene are also shown. (e) Unamplified genes in the strain panels with linear increases in expression across the panel (FDR = 0.05). Pooled set of amplified genes with expression proportionate to gene copy (from *Figure 9*). (c) Dosage compensated genes (from *Figure 9*).

### Major datasets

The following datasets were generated:

| Author(s) | Year | Dataset title | Dataset ID and/or URL | Database, license, and accessibility information |
|---|---|---|---|---|
| Hose J, Gasch A | 2014 | Dosage compensation can buffer copy-number variation in wild yeast | https://www.ncbi.nlm.nih.gov/geo/query/acc.cgi?acc=gse61532 | Publicly available at NCBI Gene Expression Omnibus (GSE61532). |
| Hose J | 2014 | Saccharomyces cerevisiae Genome Sequencing of Aneuploid Wild Strains | http://www.ncbi.nlm.nih.gov/Traces/sra/?study=SRP047341 | Publicly available at NCBI Sequence Read Archive (SRP047341). |

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

## Appendix 1

# Statistical analysis of DNA/RNA association

### 2.1 Initial Data Processing

The data analysis pipeline for each strain panel involved a series of preprocessing steps to prepare data for the mixture-model-based cluster analysis.

1. *Generation:* The sequencing facility at the UW Biotechnology Center aligned raw reads and produced read counts per yeast gene for both DNA and RNA from a number of samples in each strain panel.
2. *Coverage filter*: We removed genes for which mean (over different experimental cells) RNA counts per gene <5. This cutoff was defined empirically based on reproducibility across the strain panels.
3. *Zeros:* Where RNA or DNA count was 0, we replaced by 1/2 to avoid downstream log errors.
4. *DNA/RNA Alignment:* Genomic DNA counts were measured in biological duplicate. The replicates were averaged for each strain, and then the average normalized DNA counts per gene were used as the reference point for all three biological RNA measurements for that strain.
5. *Library size adjustment:* For each profile, we summed over all chromosome library sizes and corrected for library size. I.e., we divided counts by profile-specific (whole genome) library size.
6. *Reference normalization:* We divided each experimental profile from the aneroid strains by its associated (via replicate) euploid control profile.
7. *Log transform:* We transformed relative counts to natural logarithm scale.
8. *Aneuploid filter:* We restricted to chromosome 8 genes (for West African strain NCYC110) or to chromosome 12 genes (for the YPS panel).
9. *Nonlinear filter:* We filtered genes showing nonlinear DNA/RNA relationship on the transformed scale (see non-linear filter, below).
10. *Output:* We sent the processed data to the mixture of linear regression (MLR) clustering calculation.

### 2.2 Non-linear filter

We had 9 DNA measurements (3 repeated measures for 2n, 3n, and 4n, respectively) and 9 RNA measurements, for each gene under study and for both strain panels. From above, these measurements were on a logarithmic scale, and normalized to a reference strain. Owing to design of the controls, the DNA values were identical across the three replicate samples within each strain, and recorded changes in the relative DNA copy number. If RNA/DNA followed a linear regression line (as in **Appendix 1 Figure 1**), then,

$$\frac{B - A}{C - A} = \frac{3n - 2n}{4n - 2n}. \tag{1}$$

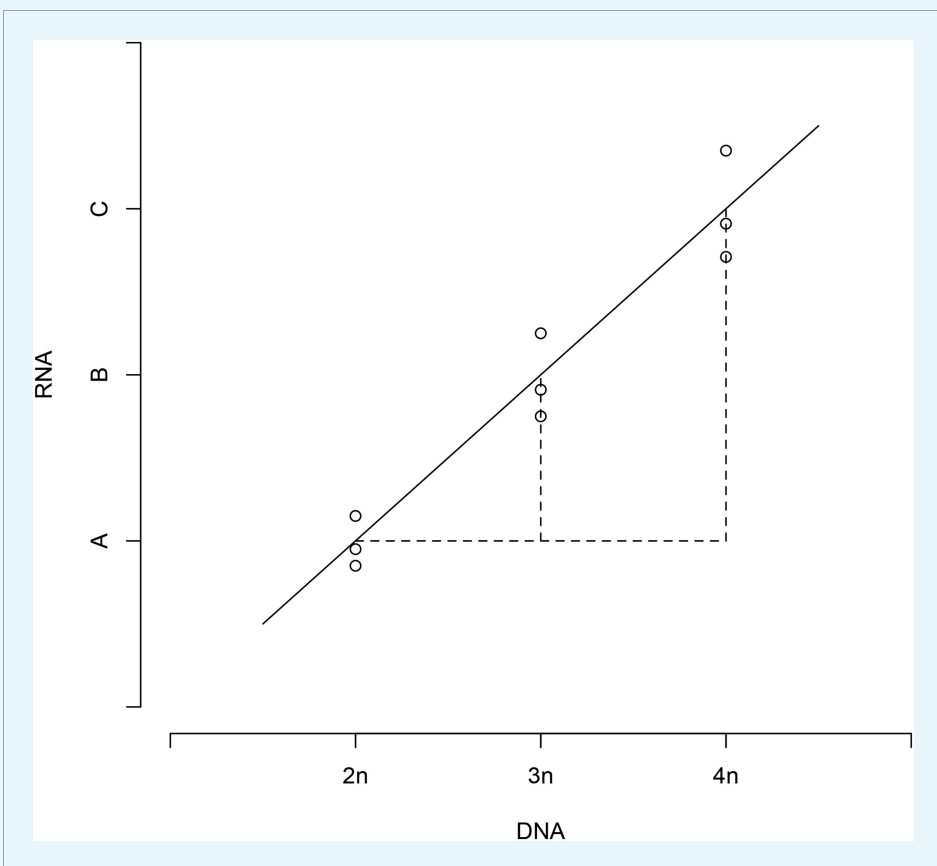

**Appendix 1 Figure 1**. Relative DNA and RNA values follow a linear regression.

A non-linear relationship would violate this rule. To test rule (1), we fit an ANOVA model:

$$\text{RNA} \sim \text{Aneuploid type (a factor with three levels : 2n, 3n, and 4n),} \qquad (2)$$

and observed that 1 is equivalent to $\dfrac{\text{coeff\_3n}}{\text{coeff\_4n}} = \dfrac{3n - 2n}{4n - 2n}$. That is, the coefficients of model 2 satisfied $(4n - 2n)\text{coeff\_3n} - (3n - 2n)\text{coeff\_4n} = 0$. We deployed the test with the help of the R package car (**Fox and Weisberg, 2011**), taking care with the order of the DNA values. The normal-theory p-value is:

$$\text{linear hypothesis (lm(RNA} \sim \text{Aneuploid type, c(0, 4n - 2n, -(3n - 2n), 0)}\backslash\$'\text{Pr}(\$>\$\text{F})'[2]))$$

Genes were flagged as possibly having a non-linear relationship if this p-value was less than 0.05.

## 2.3 Clustering by MLR

### 2.3.1 Overview

For a number of yeast genes $\{g\}$ we had measurements on the relative abundance of genomic DNA $x_g = (x_{g,i})$ and matched RNA $y_g = (y_{g,i})$ in cultured cells. Here $i$ indexes the specific measurement, running from $i = 1$ to the number of samples $n_g$ in the strain panel. Having removed genes exhibiting a non-linear DNA/RNA relationship, our goal was to cluster genes according to the pattern of their linear relationship. Phenomena related to dosage compensation may be reflected in this relationship, and by clustering genes we hoped to gain further insight into these phenomena.

We sought to cluster according to several underlying patterns: (1) intercept 0, slope 1, (2) intercept negative, slope 1, (3) slope less than one. These patterns, when considered on the relative measurement scale, refer to three basic regulatory regimes of interest. We developed a method based on treating the data as a mixture of normal components, and within a specific probability model we computed the posterior probability that each gene arose from any of these components, following the prescripts of model-based mixture clustering (**McLachlan and Peel, 2000**). An approximate EM algorithm was constructed to compute these probabilities and thus determine the clustering of genes according to their probable regime of linear DNA/RNA relationship.

## 2.3.2 Normal mixture model

Firstly, we conditioned on DNA measurements $\{x_g\}$, as is common in regression analysis. We treated the RNA vectors $\{y_g\}$ as the realization of Gaussian random vectors $\{Y_g\}$, whose mean and covariance depended on latent classes in operation for the different genes. To develop this further, we supposed,

$$Y_{g,i} = \alpha_g + \beta_g x_{g,i} + \epsilon_{g,i} \quad \text{for } i = 1, 2, \ldots, n_g, \tag{3}$$

where $\alpha_g$ and $\beta_g$ were the gene-specific intercept and slope, respectively, and where $\epsilon_{g,i}$ was a mean zero Gaussian random error that had variance $\sigma_g^2$. Five distinct hypotheses (classes, clusters) were considered:

$$1. \ \alpha_g = 0, \ \beta_g = 1,$$

$$2a. \ \alpha_g < 0, \ \beta_g = 1,$$

$$2b. \ \alpha_g > 0, \ \beta_g = 1,$$

$$3a. \ \beta_g < 1,$$

$$3b. \ \beta_g > 1,$$

Cases 2*b* and 3*b* were entertained for completeness, though they were expected to be less intepretable than the primary classes of interest: 1, 2*a*, and 3*a*. In classes 3*a* and 3*b*, the intercept was not constrained. The model was strengthened by not requiring that we fix or estimate parameters $\{\alpha_g, \beta_g\}$. Specifically, we treated these as random effects, governed by suitably constrained Gaussian distributions that respect the class structure. The random-effects distribution was as follows:

$$1. \ \alpha_g = 0, \ \beta_g = 1,$$

$$2a. \ \alpha_g \sim \text{Normal}(0, \sigma_\alpha^2) \text{ restricted to } (-\infty, 0), \ \beta_g = 1,$$

$$2b. \ \alpha_g \sim \text{Normal}(0, \sigma_\alpha^2) \text{ restricted to } (0, \infty), \ \beta_g = 1,$$

$$3a. \ \alpha_g \sim \text{Normal}(0, \sigma_\alpha^2), \ \beta_g \sim \text{Normal}\left(1, \sigma_\beta^2\right) \text{ restricted to } (-\infty, 1),$$

$$3b. \ \alpha_g \sim \text{Normal}(0, \sigma_\alpha^2), \ \beta_g \sim \text{Normal}\left(1, \sigma_\beta^2\right) \text{ restricted to } (1, \infty),$$

These class-specific distributions were governed by two variance components $\sigma_\alpha^2$ and $\sigma_\beta^2$, which we estimated from the data. They express variation within each class and across genes in the specific values of intercepts and slopes. Formally, we assumed mutual independence among all $\{\alpha_g\}$, $\{\beta_g\}$ and errors $\{\epsilon_{g,i}\}$. The mixture model was fully specified by introducing discrete latent class random variables $\{Z_g\}$ and parameters $\lambda = (\lambda_j)$, to be estimated from the data, defined,

$$P(Z_g = j) = \lambda_j \quad j \in \{1, 2a, 2b, 3a, 3b\}. \tag{4}$$

After parameter estimation, model-based clustering was based on the posterior probabilities:

$$P\left(Z_g = j | x_g, y_g\right) = \text{constant} \times \lambda_j \times p\left(y_g | Z_g = j, x_g\right), \tag{5}$$

with the constant computed by normalizing the probabilities to sum to one. The key ingredients in (5) were the class-specific probability densities $p(y_g | Z_g = j, x_g)$, which measured how well the data were explained by each class $j$. Note that $p(y_g | Z_g = j, x_g)$ is a multivariate joint density, since $y_g$ is a vector of RNA measurements, and it is also a marginal density, because the latent random effects $\alpha_g$ and $\beta_g$ were integrated away. An interesting element of the proposed approach was that the densities $p(y_g | Z_g = j, x_g)$ were available analytically, in much the same way as marginal compound Gamma distributions were computed in **Newton et al. (2004)**.

**Class 1**: there was no randomness in the intercept slope in this class, so the RNA vector $Y_g$ satisfied:

$$Y_g \sim \text{Normal}_{n_g}\left(x_g, \sigma_g^2 I\right), \tag{6}$$

where $I$ is the $n_g \times n_g$ identity matrix. Then $p(y_g | Z_g = 1, x_g)$ is the ordinate of the multivariate normal density in (6): up to a constant across classes,

$$\log p\left(y_g | Z_g = 1, x_g\right) = -\frac{n_g}{2} \log\left(\sigma_g^2\right) - \left(\frac{1}{2\sigma_g^2}\right)\left(y_g - x_g\right)^t \left(y_g - x_g\right). \tag{7}$$

**Classes 2a, 2b**: in these classes, it was convenient to introduce a related vector $\tilde{Y}_g$ based on having no constraints on $\alpha_g$; that is taking $\alpha_g \sim \text{Normal}(0, \sigma_\alpha^2)$. Without the ordering constraint, it was immediate that,

$$\tilde{Y}_g \sim \text{Normal}_{n_g}\left(x_g, \sigma_g^2 I + \sigma_\alpha^2 e e^t\right), \tag{8}$$

where $e$ is an $n_g \times 1$ vector of 1's. If $\tilde{p}_2$ denotes the density of the multivariate normal in (8), then we found:

$$p\left(y_g | Z_g = 2a, x_g\right) = \tilde{p}_2\left(y_g | x_g\right) \tilde{P}_2\left(\alpha_g < 0 | y_g, x_g\right), \tag{9}$$

$$p\left(y_g | Z_g = 2b, x_g\right) = \tilde{p}_2\left(y_g | x_g\right) \tilde{P}_2\left(\alpha_g > 0 | y_g, x_g\right).$$

Here, the probabilities involving $\tilde{P}_2$ refer to conditional probabilities about the random intercept in the *unconstrained* model for $\alpha_g$. From standard conjugate Bayesian analysis, the unconstrained posterior for $\alpha_g$ was,

$$\alpha_g | y_g, x_g \sim \text{Normal}\left(\frac{e^t\left(y_g - x_g\right)}{\sigma_g^2\left(\frac{1}{\sigma_\alpha^2} + \frac{n_g}{\sigma_g^2}\right)}, \frac{1}{\frac{1}{\sigma_\alpha^2} + \frac{n_g}{\sigma_g^2}}\right), \tag{10}$$

and so the ordering factors on the right side of (9) were readily computed from univariate Gaussian CDF's. As to the unconstrained ordinate $\tilde{p}_2$, following (7),

$$\log \tilde{p}_2\left(y_g | x_g\right) = -\frac{1}{2}\log \det \Sigma_2 - \frac{1}{2}\left(y_g - x_g\right)^t \Sigma_2^{-1}\left(y_g - x_g\right), \tag{11}$$

where $\Sigma_2 = \sigma_g^2 I + \sigma_\alpha^2 e e^t$ as in (8). Using Woodbury's formula, this log density was readily computed in terms of summary statistics from gene $g$.

**Classes 3a, 3b**: here, the constraints were on the slopes $\beta_g$, but randomness in both slopes and intercepts must be accounted for. Again we computed ordinates by considering a model in which there were no constraints on $\beta_g$. (Call it $\tilde{p}_3$.) In comparison with (8),

$$\tilde{Y}_g \sim \text{Normal}_{n_g}\left(x_g, \Sigma_3 = \sigma_g^2 I + \sigma_\alpha^2 ee^t + \sigma_\beta^2 x_g x_g^t\right). \tag{12}$$

To compute $p(y_g|Z_g = 3a, x_g)$ (for example), we required the normal mass $\tilde{P}_3(\beta < 1|x_g, y_g)$ from the unconstrained model as well as the unconstrained ordinate $p_3(y_g|x_g)$, which was,

$$\log \tilde{p}_3\left(y_g|x_g\right) = -\frac{1}{2}\log \det \Sigma_3 - \frac{1}{2}\left(y_g - x_g\right)^t \Sigma_3^{-1}\left(y_g - x_g\right). \tag{13}$$

Convenient formulas for these quantities were available in terms of gene-level statistics.

