## [Decision Letter]

Thank you for sending your work entitled “Dosage compensation buffers copy-number variation in wild yeast” for consideration at *eLife*. Your article has been favorably evaluated by Stylianos Antonarakis (Senior editor), Duncan Odom (Reviewing editor), and three reviewers. The Reviewing editor and the reviewers discussed their comments before we reached this decision, and the Reviewing editor has assembled the following comments to help you prepare a revised submission.

The consensus was that the demonstration that wild yeast strains exhibit gene expression compensation in the face of aneuploidy was of potentially high interest to a broad community. Some substantial criticisms were raised, in particular by Reviewers 1 and 2, which you must carefully address before this can be published in *eLife*.

The essential revision requirements are: (1) repeat the chromosome titrations you did for Chr12 in strain YPS1009 using the Chr8 strains derived from NCYC110, and this new analysis must include spike-in controls for RNA-seq; (2) much better annotation of the strains used both in main text and in the supplemental.

Like the three reviewers have done, the Reviewing Editor, Duncan Odom, has gone over your manuscript both editorially and graphic design-wise, and we have made suggestions below that should help improve your clarity and presentation. One extremely important point: you must re-tool your figures with an eye towards red–green color blindness.

Reviewer #1:

This is a very interesting paper looking at aneuploidy in non-lab strains of yeast. They conclude that natural isolates are more tolerant of aneuploidy than the lab strain W303 and they suggest that this is related to an apparent dosage compensation of gene expression from the amplified chromosome.

I think this is important work, and it is getting to the heart of several key conundrums in the aneuploidy field. Because it's such an important problem, I think the paper should do more to convincingly show the results and how they contrast with the lab strain results. I suspect, as the authors find in one of the RNAseq experiments, that their results are not actually as different as it first seems.

1) The first set of RNAseq experiments are problematic for a few reasons. First, they are not performed with isogenic strains that only differ by presence/absence of a CNV. The additional variation in gene expression introduced by all the other genetic differences going on could be partly responsible for the observed differences. I think it is crucial to also show at least a few paired euploid samples of similar relatedness to demonstrate the degree of expression difference expected in the absence of any CNVs. The reason this is important can be best explained by imagining that the base expression really does change proportional to the copy number. Now imagine additional genetic differences cause many genes to change in expression by, say 1.5-fold up or down, a result that has frequently been observed in comparisons between natural isolates. In the simplest possible additive model of expression, many genes in the CNV would now show greater or less than expected changes in expression, but not due to dosage compensation.

Also, I think the data from these experiments would be much easier to interpret if displayed as histograms (e.g. something like Figure 2 in the recent *eLife* paper from Amon and Torres) in addition to the scatter plot view. Furthermore, the analysis would be much more compelling if compared directly to data from at least the W303 gene expression series, and preferably additional aneuploid lab strains (for example, the Hughes et al. deletion collection strains that are aneuploid, or some of the aneuploids isolated from experimental evolution). Though on average genes in the CNV change proportionately with copy number in these lab strain experiments, that is certainly not universally true, nor should it be: the W303 experiments, for example, have the same problem as described above, that is, the layering of multiple expression signatures, in this case the stress and/or growth rate response coupled with the copy number driven expression. Evolved strains and deletion strains have additional mutations as well that may affect expression on top of the perturbation from the CNV. Direct comparison to these data would be very convincing if there really is a big difference here. It certainly wouldn't be the first time that lab strains are outliers in behavior, so I am inclined to believe there is something to see, but it should be shown explicitly.

2) Some of the concerns about the first set of experiments are mitigated by the dosage series generated for the second batch of RNAseq. Here, we have both isogenic lines carrying different numbers of chromosomes, and a direct comparison with the lab strain. The fact that the apparent dosage compensation is present in the W303 3N strain in this experiment is one thing that makes me suspect these results are not as different as initially posited (and see the paragraph above for ways of demonstrating that I'm wrong). Also, that the 4N W303 strain is the aberrant one, and grows most poorly, hints again at the issue with confounding additional gene expression signatures.

Additionally, I really wish that chromosome 12 had not been the core element of this part of the paper. The rDNA array is on chromosome 12, and differences in repeat number can change the size of the chromosome by up to a megabase. In the W303 experiments, chromosome 12 disomy causes a growth defect that I definitely would agree fits the definition of “extreme.” Also problematic, different strains carry polymorphisms in the rDNA that cause all sorts of potentially confounding phenotypes, such as issues with replication (see Kwan et al. 2013 PLoS Genetics), a problem that might be particularly important for aneuploids. I think the experiments are still interesting, of course, but they are difficult to interpret without some indication that the rDNA is similar in sequence and copy number in the different strain backgrounds. A chromosome 8 series in W303 (or at least one other matched set) would also be helpful in addressing the generalizability of the observations with 12.

3) The remaining parts of the paper are nice follow-up experiments, including the analysis over other strains of genes that are found in CNVs more or less frequently, and the plasmid expression experiments. I have no major comments about these analyses and appreciated their inclusion.

Reviewer #2:

The Hose et al. manuscript examines the extent of dosage compensation within wild yeast strains. The work examines RNA-seq data and uses a new model (mixture of linear regression) to identify distinct classes of dosage compensation. They nicely use existing publicly available data to the level of dosage compensation constraint (Vg/Vm) and CNV buffering. In the data leading up to these sections, however, there are some major concerns.

In general differential expression studies using RNA-seq (subsection headed “A common aneuploidy response recapitulates a Down syndrome signature”; and edgeR as described here) assume that most things are not changing, an assumption that could very well be violated by whole chromosome amplifications. Normalization to a closely related euploid would not alleviate this issue. Spike in controls are necessary in this case. Redoing all their RNA-seq studies is unnecessary if spike in normalization for one strain relative to its closely related euploid shows largely identical results.

In the subsection headed “Many amplified genes display lower-than-expected expression”, it is noted that strains are “isogenic aside of the aneuploidy” yet this assertion is made without proof. In particular, the construction of the aneuploid isogenic panel for NCYC110 required passaging for 427 generations. At the known yeast mutation rate, this would result in roughly 2-3 mutations pre haploid genome content (or as many as 12 mutations in their strain as it is tetraploid). As anueploidy tolerating mutations are known to arise quickly (Torres et al. Cell 2010), it is unclear whether the observed differences arise from changes in aneuploidy or mutations. Given the diploid arose quickly after the triploid (17 generations), is the triploid strain unstable?

Given ambiguity in Figure 1 and the strain table, it is unclear if these were strains they already genome sequenced (in which case this would be a bioinformatic exercise) or whether genome sequencing is necessary.

In general the base ploidy and nature of the aneuploidy within various Figures is unclear. Given the broad range of comparisons made within this paper (strain backgrounds, aneuploidies, ploidies, etc), it is imperative for interpretation that these be more clearly labeled and identified. Likewise, their strain table should indicate for which strains genome sequence and RNA-seq are available from this study. It is unclear how “closely related euploids as paired controls” was determined. How do you know a particular strain is closely related: by genome sequence, by lineage, other?

Reviewer #3:

The authors observed no difference in growth rates or doubling times in WT induced aneuploidy strains, suggesting no deleterious effects are derived from this. Moreover, aneuploidy was stable throughout generations, strengthening the hypothesis that WT strains tolerate aneuploidy.

The authors elegantly generated aneuploidy strains from euploid parents, which give strength to their conclusions and showed dosage compensation is one mechanisms involved in such tolerability generating less than expected gene expression in some of the amplified genes that are under higher evolutionary constrain due to their toxic effects if overexpressed.

Comments:

The observation of downregulation of respiration-related genes is very interesting, yet to conclude that induced aneuploidy in WT strains recapitulates a Down syndrome signature is not sustained by the data shown in this manuscript. I would leave this assumption/interesting correlation for discussion and only highlight mitochondrial ribosomal proteins and respiration genes signature.

Do the authors have a hypothesis on how this signature is involved in the observed permissive aneuploidy?

Is Figure 4 showing the overall number of lower and higher than expected regulated genes in all aneuploid cells versus their euploid? How many of those are common amongst all strains tested? How would this graph look for each individual strain aneuploidy/euploid comparison?

Duncan Odom, Reviewing editor:

My comments are focused on presentational and editorial issues.

The Title should read: “Dosage compensation can buffer copy-number variation in wild yeast”.

The Abstract should be carefully re-edited, as it has a few awkward sentences.

In the Introduction, what criteria were used in order to pair up your yeast samples was not at all clear. This is mentioned in the major comments numbered above, but this section will need heavy and careful revision.

In the Results section, the use of these seemingly arbitrary classes is confusing to the reader. Consider how else this section could be more clearly presented. Related: the number of genes in each class should be listed clearly in the text.

In the Discussion, the term 'Balance Hypothesis' requires both careful explanation (I have no idea what this is, for instance) and a few accessible references or reviews.

In the Discussion, you state: “while the expression of these genes is controlled…” Which genes? This is a very ambiguous and poorly structured statement.

Figures: In general, please consider re-tooling all figures using the principles outlined in Visual Display of Quantitative Information by Edward Tufte. However, below I highlight specific examples that should be corrected.

The use of colors in the figures is poorly considered throughout. Random color choices appear to occur in their bar charts. There is excessive use of bar charts, which reduce interest.

Required: Figure 2 and Figure 2 must not use red-green, as this is impossible for the 6-8% of male readers to see who are colorblind. Replace with yellow-blue instead. See:http://www.nature.com/nmeth/journal/v8/n6/full/nmeth.1618.html

Figure 4 is poorly presented, and the axes are not understandable. Consider breaking into two separate and better annotated scatterplots.

Figure 6. D panel is not informative. What is this trying to say?

Figure 7. RNA abundance used blue in A/B and red in C/D. Why?

---

## [Author Response]

*The essential revision requirements are: (1) repeat the chromosome titrations you did for Chr12 in strain YPS1009 using the Chr8 strains derived from NCYC110, and this new analysis must include spike-in controls for RNA-seq; (2) much better annotation of the strains used both in main text and in the supplemental*.

We have added 16 new RNA-seq samples to our analyses, all done with cell- based spike-in *Sz. pombe* controls. As described in more detail in the response to Reviewer #2, we performed several different analyses to show that the spike-in normalized data are virtually indistinguishable from the RPKM normalized data, although RPKM normalization was more reproducible across biological replicates. This justifies the use of RPKM normalization used in this study.

We also analyzed dosage-compensated genes in the spike-in normalized NCYC110_Chr8 strain panel and show that the results recapitulate the results reported in the original analysis. Together, these analyses show that our main conclusion—that some amplified genes are subject to dosage compensation in naturally aneuploid yeast—is not an artifact of RNA-seq normalization.

*One extremely important point: you must re-tool your figures with an eye towards red-green color-blindness*.

The figure colors, especially the transcriptome heat-map images in Figure 2, have been updated as requested, using a blue-yellow color scale.

Reviewer #1:

*1) The first set of RNAseq experiments are problematic for a few reasons. First, they are not performed with isogenic strains that only differ by presence/absence of a CNV. The additional variation in gene expression introduced by all the other genetic differences going on could be partly responsible for the observed differences. I think it is crucial to also show at least a few paired euploid samples of similar relatedness to demonstrate the degree of expression difference expected in the absence of any CNVs. The reason this is important can be best explained by imagining that the base expression really does change proportional to the copy number. Now imagine additional genetic differences cause many genes to change in expression by, say 1.5-fold up or down, a result that has frequently been observed in comparisons between natural isolates. In the simplest possible additive model of expression, many genes in the CNV would now show greater or less than expected changes in expression, but not due to dosage compensation*.

First, we thank the reviewer for their positive assessment of the work. The reviewer is absolutely correct that other genetic differences can contribute to the expression differences between aneuploid strains and their paired euploid references. This is the main motivation for the strain panels that we focused on in the paper, which do not suffer from this limitation.

However, to fully address the reviewer’s concern, we have now added additional sequencing data for three aneuploid strains (two natural aneuploids and one derived aneuploid) and their isogenic euploid partners. The strains in each pair were derived from one another and differ only in the number of affected chromosomes (as supported by SNP analysis in the paired strains). The addition of this analysis allowed us to substantially expand the number of dosage-compensated genes identified, bolstering in the statistical power throughout the paper. The new analysis was done with spike-in normalization controls, which gives virtually the same data as RPKM normalization.

*Also, I think the data from these experiments would be much easier to interpret if displayed as histograms (e.g. something like*
Figure 2
*in the recent* eLife *paper from Amon and Torres) in addition to the scatter plot view. Furthermore, the analysis would be much more compelling if compared directly to data from at least the W303 gene expression series, and preferably additional aneuploid lab strains (for example, the Hughes et al. deletion collection strains that are aneuploid, or some of the aneuploids isolated from experimental evolution). Though on average genes in the CNV change proportionately with copy number in these lab strain experiments, that is certainly not universally true, nor should it be: the W303 experiments, for example, have the same problem as described above, that is, the layering of multiple expression signatures, in this case the stress and/or growth rate response coupled with the copy number driven expression. Evolved strains and deletion strains have additional mutations as well that may affect expression on top of the perturbation from the CNV. Direct comparison to these data would be very convincing if there really is a big difference here. It certainly wouldn't be the first time that lab strains are outliers in behavior, so I am inclined to believe there is something to see, but it should be shown explicitly*.

Expression in the W303_Chr12 strain panel indicates that the strain has massive secondary stress responses to the aneuploidy; this clearly alters expression of many amplified genes (driving some lower and some higher in expression due to the stress response). Because of the completely aberrant response of this strain, it is not appropriate to use the strain as a reference point. We now provide data for five different strains backgrounds for which we have paired aneuploid-euploid strains or strain panels, with few identified SNPs; furthermore, any SNPs are unlikely to be common across the five strain groups. Thus, the simplest explanation is that our results are reflecting on dosage compensation in natural strains.

It is likely true that the W303_Chr12-3n strain utilizes some form(s) of dosage compensation at specific genes, and in fact our results support this conclusion. We have clarified this point in the main text (in subsection headed “Dosage compensation is mediated by multiple mechanisms”). Because the manuscript now covers a large number of different aneuploid strains, we have opted not to show individual histograms for each strain but rather focus on pooled analysis of the genes across strains.

*2) Some of the concerns about the first set of experiments are mitigated by the dosage series generated for the second batch of RNAseq. Here, we have both isogenic lines carrying different numbers of chromosomes, and a direct comparison with the lab strain. The fact that the apparent dosage compensation is present in the W303 3N strain in this experiment is one thing that makes me suspect these results are not as different as initially posited (and see the paragraph above for ways of demonstrating that I'm wrong). Also, that the 4N W303 strain is the aberrant one, and grows most poorly, hints again at the issue with confounding additional gene expression signatures*.

*Additionally, I really wish that chromosome 12 had not been the core element of this part of the paper. The rDNA array is on chromosome 12, and differences in repeat number can change the size of the chromosome by up to a megabase. In the W303 experiments, chromosome 12 disomy causes a growth defect that I definitely would agree fits the definition of “extreme.” Also problematic, different strains carry polymorphisms in the rDNA that cause all sorts of potentially confounding phenotypes, such as issues with replication (see Kwan et al. 2013 PLoS Genetics), a problem that might be particularly important for aneuploids. I think the experiments are still interesting, of course, but they are difficult to interpret without some indication that the rDNA is similar in sequence and copy number in the different strain backgrounds. A chromosome 8 series in W303 (or at least one other matched set) would also be helpful in addressing the generalizability of the observations with 12*.

We have scored the rDNA repeats based on read counts from genomic DNA sequencing across the YPS1009_Chr12 strain panel: the depth of coverage indicates that rDNA reads increase as expected across the strain panel, proportionate to the increase in Chr 12 copy across. Thus, we do not believe that gross differences in rDNA repeat number affects the tolerance to aneuploidy in this strain. Incidentally, Chr 12 amplification in W303 was reported to be among the least toxic aneuploidy in that strain background (Sheltzer et al*.* PNAS 2012). With the newly added data, we now have five isogenic aneuploid-euploid strain groups spanning three chromosomes (Chr 12, Chr 8, and Chr 16). Thus, our results cannot be explained by differences in rDNA copy number on the amplified chromosome.

Reviewer #2:

*In general differential expression studies using RNA-seq (subsection headed “A common aneuploidy response recapitulates a Down syndrome signature“; and edgeR as described here) assume that most things are not changing, an assumption that could very well be violated by whole chromosome amplifications. Normalization to a closely related euploid would not alleviate this issue. Spike in controls are necessary in this case. Redoing all their RNA-seq studies is unnecessary if spike in normalization for one strain relative to its closely related euploid shows largely identical results*.

As requested by this reviewer and the editors, we have redone some of the RNA-seq analysis with spike-in *Schizosaccharomyces pombe* controls (including a biological replicate of the entire NCYC110 strain panel as required by the editor and duplicated aneuploid-euploid strain pairs for three new strains). The bottom line is that the spike-in normalization gives data that are virtually indistinguishable from RPKM normalized data, and the dosage-compensation phenotype we previously reported in the NCYC110_Chr8 strain panel is clearly recapitulated in the new replicates normalized by spike-in controls.

To summarize these experiments:

The spike-in approach we use involves mixing a precise number of *Sz. pombe* cells (grown in a single large-scale batch) with each *S. cerevisiae* strain under consideration. A critical aspect is that the *Sz. pombe:S. cerevisiae* cell ratio must be identical across all strains and samples.

This is extremely challenging to do with wild strains, many of which are flocculent and thus impossible to count accurately. We went to extreme lengths to carefully estimate cell numbers for mixing purposes. Because mixing is done on a cell-count basis before RNA isolation, library preps, and bulk sequencing, *S. cerevisiae* reads across samples can be normalized by setting the slope of *Sz. pombe* read counts across samples to 1.0.

We show that the results are virtually the same as RPKM normalization, through three approaches. First, we compared the slope across biological replicates that were normalized with each of the two methods; because these are replicates, the slope across all genes is expected to be 1.0. Indeed, the average slope across replicates for spike-in normalization (1.052 +/- 0.042) was very similar to the average slope across replicates normalized by RPKM (0.992 +/- 0.026). The slope when comparing spike-in normalized data to RPKM normalized data was effectively 1.0, indicating that the values across the transcriptome were nearly identical when normalized with the two approaches. Second, we compared the slope across the transcriptome when comparing aneuploid strains to their isogenic euploid partners: the average slope is essentially the same (1.07 or 1.01) when the data were normalized by spike-in controls versus RPKM, although the standard deviation was significantly lower for RPKM (0.019) compared to spike-in controls (0.15), likely owing to the challenges with cell -based spiking. Third, we used the replicated NCYC110_Chr8 strain panel data to assess the fit between the mRNA and DNA abundance across NCYC110_Chr8 cells carrying two, three, or four copies of Chr 8. Recall that for a given gene, an mRNA *versus* DNA slope <1.0 is indicative of dosage compensation, whereas cells that escape compensation display a slope of the mRNA-DNA fit of 1.0 (or in some cases higher). The mRNA-DNA slopes across the panel are virtually indistinguishable when the data are normalized by RPKM versus spike-in controls (slope of the slopes = 0.9997). This indicates that the mRNA-versus-DNA fits across the dataset are the same whether the data are normalized by spike-in controls or RPKM.

Finally, in accordance with the editor’s request, we reassessed the mRNA-DNA fit across the new NCYC110_Chr8 strain panel data normalized by the spike-in controls, compared to the strain-panel data normalized by RPKM as reported in the original manuscript. Of the 27 genes previously classified as dosage-compensated (Class 3a), 24 (89%) have reduced mRNA-DNA slopes (<1.0) in the newly added spike-in normalized NCYC110_Chr8 strain panel – thus the new data fully support our original analysis.

The Methods section has been updated to describe the spike-in controls and to briefly summarize our conclusions that the two normalization methods give essentially the same data. This analysis justifies our usage of RPKM throughout the paper.

In the subsection headed “Many amplified genes display lower-than-expected expression”, it is noted that strains are “isogenic aside of the aneuploidy” yet this assertion is made without proof. In particular, the construction of the aneuploid isogenic panel for NCYC110 required passaging for 427 generations. At the known yeast mutation rate, this would result in roughly 2-3 mutations pre haploid genome content (or as many as 12 mutations in their strain as it is tetraploid). As anueploidy tolerating mutations are known to arise quickly (Torres et al. Cell 2010), it is unclear whether the observed differences arise from changes in aneuploidy or mutations. Given the diploid arose quickly after the triploid (17 generations), is the triploid strain unstable?

We used GATK to call SNPs across the strain panel, focusing on SNPs identified in both replicates of the genomic DNA sequencing (for the replicated data in the strain panels). We identified only one homozygous SNP in the NCYC110_Chr8-3n strain and a different homozygous SNP in the NCYC110_Chr8-2n strain, which were identified in both of the replicate genomic DNA sequences. We added a short description of this analysis to the Methods and clarify that most of the effects we see are likely to be due to the aneuploidy. In the case of genes amplified on Chr 8, we now have independent data across three different strain panels. The strong agreement across strains indicates that the results we see are not due to mutations in the selected strains, which are very unlikely to have picked up the same mutations.

We point out that while mutations to tolerate aneuploidy are quickly selected in lab studies (driven by the strong selective pressure to overtake sickly aneuploids), here we are selecting for the loss of aneuploidy, which provides a mild fitness benefit. The triploid NCYC110_Chr8-3n strain is in fact no less stable than the tetraploid or euploid and was isolated by chance after only 17 generations (since triploids took over the population later in the experiment). Control experiments since our original submission confirm that the trisomic strain is very stable.

*Given ambiguity in*
Figure 1
*and the strain table, it is unclear if these were strains they already genome sequenced (in which case this would be a bioinformatic exercise) or whether genome sequencing is necessary*.

We apologize for the confusion. All of the genomes and RNA-seq discussed here were generated as part of this project.

*In general the base ploidy and nature of the aneuploidy within various Figures is unclear. Given the broad range of comparisons made within this paper (strain backgrounds, aneuploidies, ploidies, etc), it is imperative for interpretation that these be more clearly labeled and identified. Likewise, their strain table should indicate for which strains genome sequence and RNA-seq are available from this study. It is unclear how “closely related euploids as paired controls” was determined. How do you know a particular strain is closely related: by genome sequence, by lineage*, *other?*

We have clarified the strain naming convention to list strain name followed by the chromosome and aneuploidy state (e.g. YPS1009_Chr12-3n indicates that strain YPS1009 carries three copies of Chr 12). In most cases here, strains are of the diploid ploidy state (with the exception of some controls in Figure 1 and Figure 7). We have clarified the naming convention in several places in the main text and updated the strain table ([Supplementary-material SD1-data]). We have also added a statement to the main text and the Methods explaining how paired euploid controls were chosen: if known in advanced, the paired euploid was chosen as the most closely genetically related strain, based on phylogenetic analysis, but in two cases where we did not have a priori information we chose strains from the same ecological group.

Reviewer #3:

*The observation of downregulation of respiration-related genes is very interesting, yet to conclude that induced aneuploidy in WT strains recapitulates a Down syndrome signature is not sustained by the data shown in this manuscript. I would leave this assumption/interesting correlation for discussion and only highlight mitochondrial ribosomal proteins and respiration genes signature*.

Do the authors have a hypothesis on how this signature is involved in the observed permissive aneuploidy?

We have moved these points to a short mention in the Discussion. At this point, we do not understand why this response occurs in aneuploid strains but it is something we are pursuing in the lab.

*Is*
Figure 4
*showing the overall number of lower and higher than expected regulated genes in all aneuploid cells versus their euploid? How many of those are common amongst all strains tested? How would this graph look for each individual strain aneuploidy/euploid comparison?*

Figure 4 and newly added Figure 4 show genes with lower-than, higher-than, or as expected expression in the combined group of aneuploid strains versus euploid references. Because we have so many different strains pairs, we opted not to show individual graphs for each strain. However, with the new data added in the revised manuscript, we now have RNA-seq data for three different strain backgrounds with Chr 8 aneuploidy. Of the genes with lower-than-expected expression in either or both of the new isogenic aneuploid-euploid strain pairs, 79% showed lower-than-expected expression in the NCYC110 strain panel. Although there was significant enrichment for dosage-compensated genes from the MLR model (Class 3a, *p* = 7e-3), a surprising number of genes called dosage compensated in the paired analysis were classified as having heritably reduced expression without dosage compensation (Class 2a) in the NCYC110_Chr8 panel. These results raise the possibility that there exist strain-specific differences in the genes that are subject to dosage compensation, a topic we briefly discuss in a newly added paragraph in the Discussion.

*Duncan Odom, Reviewing editor*:

*My comments are focused on presentational and editorial issues*.

*The Title should read*: *“Dosage compensation can buffer copy-number variation in wild yeast”.*

Changed accordingly.

*The Abstract should be carefully re-edited, as it has a few awkward sentences*.

Changed accordingly.

*In the Introduction, what criteria were used in order to pair up your yeast samples was not at all clear. This is mentioned in the major comments numbered above, but this section will need heavy and careful revision*.

This has been explicitly outlined in the Methods, as outlined above.

*In the Results section, the use of these seemingly arbitrary classes is confusing to the reader. Consider how else this section could be more clearly presented. Related: the number of genes in each class should be listed clearly in the text*.

The text has been extensively rewritten to more clearly describe the MLR analysis and gene classification. This presentation gives more explicit description and citation to the genes in each class, which are also listed in Table 1.

*In the Discussion, the term 'Balance Hypothesis' requires both careful explanation (I have no idea what this is, for instance) and a few accessible references or reviews*.

The statement has been clarified to: “A second prediction is posited by the Balance Hypothesis (Birchler, 2007), which asserts that expression of multi-subunit protein complexes may be more tightly controlled to maintain protein stoichiometry.”

*In the Discussion, you state: “while the expression of these genes is controlled…” Which genes? This is a very ambiguous and poorly structured statement*.

The statement has been changed to: “While the expression of dosage-compensated genes is controlled…”.

*Figures: In general, please consider re-tooling all figures using the principles outlined in Visual Display of Quantitative Information by Edward Tufte. However, below I highlight specific examples that should be corrected*.

*The use of colors in the figures is poorly considered throughout. Random color choices appear to occur in their bar charts*.

Many of the figures were designed for a supplement but integrated with the manuscript at submission to comply with the *eLife* policy on supplements. The color scheme has been changed throughout to provide a consistent palette.

*There is excessive use of bar charts, which reduce interest*.

We tried to find a balance between box plots, linear plots, and heat maps, which are used throughout. We use bar charts for growth rates and qPCR quantification. We have streamlined Figure 7 to remove several unnecessary bar charts.

*Required:*
Figure 2
*and*
Figure 2
*must not use red-green, as this is impossible for the 6-8% of male readers to see who are colorblind*. *Replace with yellow-blue instead. See:*http://www.nature.com/nmeth/journal/v8/n6/full/nmeth.1618.html

Figure 2 has been regenerated with a blue-yellow color scheme as requested.

Figure 4
*is poorly presented, and the axes are not understandable. Consider breaking into two separate and better annotated scatterplots*.

We have significantly updated Figure 4, including relabeling the axes, changing the color scheme, and specifying three (rather than two) classes of genes. The updated Figure 4 now contains a new panel analyzing the new isogenic aneuploid-euploid strain pairs.

Figure 6. *D panel is not informative. What is this trying to say?*

We feel this panel is important because it shows the full distribution of effects for all genes in the class. At this stage we have chosen to leave it as part of the manuscript.

Figure 7. *RNA abundance used blue in A/B and red in C/D. Why?*

The plot has been simplified to use one color scheme and has been streamlined to minimize the number of bar graphs.